evolution/genetics

immunogenetics, emerging infectious diseases, testudines, supertype, major histocompatibility complex, fibropapillomatosis

**Author for correspondence:**
Anna E. Savage
e-mail: anna.savage@ucf.edu

# Adaptive evolution of major histocompatibility complex class I immune genes and disease associations in coastal juvenile sea turtles

Katherine R. Martin, Katherine L. Mansfield and Anna E. Savage

Department of Biology, University of Central Florida, 4110 Libra Drive, Orlando, FL 32816, USA

  KRM, 0000-0002-1634-3484; KLM, 0000-0002-6568-2861; AES, 0000-0002-4917-8358

Characterizing polymorphism at the major histocompatibility complex (MHC) genes is key to understanding the vertebrate immune response to disease. Despite being globally afflicted by the infectious tumour disease fibropapillomatosis (FP), immunogenetic variation in sea turtles is minimally explored. We sequenced the $\alpha_1$ peptide-binding region of MHC class I genes (162 bp) from 268 juvenile green (*Chelonia mydas*) and 88 loggerhead (*Caretta caretta*) sea turtles in Florida, USA. We recovered extensive variation (116 alleles) and trans-species polymorphism. Supertyping analysis uncovered three functional MHC supertypes corresponding to the three well-supported clades in the phylogeny. We found significant evidence of positive selection at seven amino acid sites in the class I exon. Random forest modelling and risk ratio analysis of *Ch. mydas* alleles uncovered one allele weakly associated with smooth FP tumour texture, which may be associated with disease outcome. Our study represents the first characterization of MHC class I diversity in *Ch. mydas* and the largest sample of sea turtles used to date in any study of adaptive genetic variation, revealing tremendous genetic variation and high adaptive potential to viral pathogen threats. The novel associations we identified between MHC diversity and FP outcomes in sea turtles further highlight the importance of evaluating genetic predictors of disease, including MHC and other functional markers.

# 1. Introduction

Infectious diseases threaten the conservation of biodiversity, particularly in species already at risk due to human-mediated

environmental change [1,2]. Environmental change such as warming temperatures and habitat fragmentation alters interactions between hosts and pathogens [3], often increasing pathogen load [4], altering host immune function [5] and leading to altered disease risk [6]. Emerging disease in ectothermic vertebrates is especially concerning in light of their sensitivity to global change [7]. In particular, variance in environmental thermal optima is linked to altered immune function in herpetofauna [8,9], and amphibians and reptiles are host to numerous emerging infectious diseases [10,11]. How well reptiles and amphibians can respond to novel disease regimes remains an important area of investigation for conservation biology. Species may adapt to pathogen pressure through immunogenetic evolution, particularly in major histocompatibility complex (MHC) genes [12]. Thus, the extent to which immunogenetic variation modulates disease outcomes in reptiles is a critical component for understanding their ability to persist, and remains an important and open question in wildlife conservation biology.

The MHC is a genomic region in jawed vertebrates encoding proteins that recognize and present foreign peptides to T-cells to initiate acquired immune responses [13]. Classical class I and II MHC genes encode glycoproteins with highly variable peptide-binding regions (PBRs) containing peptide-binding residues that bind antigens, thus determining whether targeted pathogen removal can be achieved [14]. The MHC class I PBR derives from $\alpha_1$ and $\alpha_2$ subunits encoded by exon 2 and exon 3, respectively [15]. MHC class I proteins are typically expressed endogenously and present viral pathogen particles, while class II proteins are only expressed on antigen-presenting cells and primarily recognize extracellular pathogens [14]. The diversity of expressed MHC proteins may indicate the host's ability to respond to a suite of different pathogens, with higher MHC variation typically associated with increased pathogen resistance [12,16]. MHC diversity is linked to pathogen immunity in numerous vertebrates [17–20] and is an important indicator of overall population health.

Sea turtles are long-lived, migratory and ancient reptiles that are globally afflicted by fibropapillomatosis (FP), an infectious disease that presents as internal and external neoplastic tumours [21]. FP is associated with chelonid alphaherpesvirus 5 (ChHV5) infection [22], but causal relationships with FP development remain unclear [23]. FP most commonly affects green sea turtles, *Chelonia mydas*, but is documented in all species of sea turtles [21,24–29] and predominantly affects immature turtles in coastal habitats [21] who have recruited to neritic waters after spending their first developmental years in the open ocean [30]. Juvenile *Ch. mydas* in the southeastern United States demonstrate high FP prevalence [31], including those found on the Florida Atlantic coast [32–34]. FP consequences range from complete tumour regression and recovery to turtle mortality [35,36]. Immunosuppression is associated with FP [37] and may develop after FP onset [26,38,39]. While the prevalence of FP in coastal habitats has led to speculation that environmental cofactors such as chemical pollutants and degraded environmental quality contribute to disease development [40,41], their precise role in FP aetiology remains uncertain. The interplay of FP, environmental change, pathogens and host immune defences has remained largely uncharacterized until recently, with studies indicating reduced or altered immune function in turtles with FP [39] and in turtles in degraded habitat [42], as well as differential immune expression in FP-afflicted *Ch. mydas* [43–46]. Given the importance of MHC in the acquired immune response, characterizing MHC diversity in coastal juvenile sea turtles may be particularly informative for understanding FP susceptibility.

To date, MHC studies in reptiles have mainly characterized gene diversity and evolutionary history, finding evidence of shared alleles between closely related species, high levels of polymorphism and evidence of positive selection in antigen-binding sites [47–49]. Even within reptiles, few immunogenetic studies examine turtles [50,51], and only one study has investigated MHC in sea turtles, describing two allele lineages recovered in a single loggerhead sea turtle (*Caretta caretta*) nesting population [52]. Here, we advance knowledge of MHC genetic variation and adaptive evolution in sea turtles by sampling juvenile *Ca. caretta* from a foraging aggregate and characterizing MHC in *Ch. mydas* for the first time. Quantifying MHC repertoires from both *Ca. caretta* and *Ch. mydas* lends insight into how MHC has evolved in sea turtles and more broadly facilitates our understanding of reptilian acquired immune systems. While a relationship between pathogen susceptibility and MHC genetic variation has been documented in tortoises [53,54], associations between MHC and disease in sea turtles remain unexplored. By sampling immature *Ch. mydas*, which are more commonly affected by FP than other species or age classes [31], we are thus able to explore potential associations between MHC and FP for the first time. The University of Central Florida Marine Turtle Research Group (MTRG) has sampled juvenile *Ch. mydas* and *Ca. caretta* for over 30 years at three sites on the Atlantic coast of Florida: the Indian River Lagoon (IRL), Trident Basin (TB) and Indian River County sabellariid worm rock reef (figure 1). Mean *Ch. mydas* FP prevalence is 48.9% in the IRL [33,34], 4.8% in TB [33] and 19% in the reef [33]. *Caretta caretta* FP prevalence is 4.5%

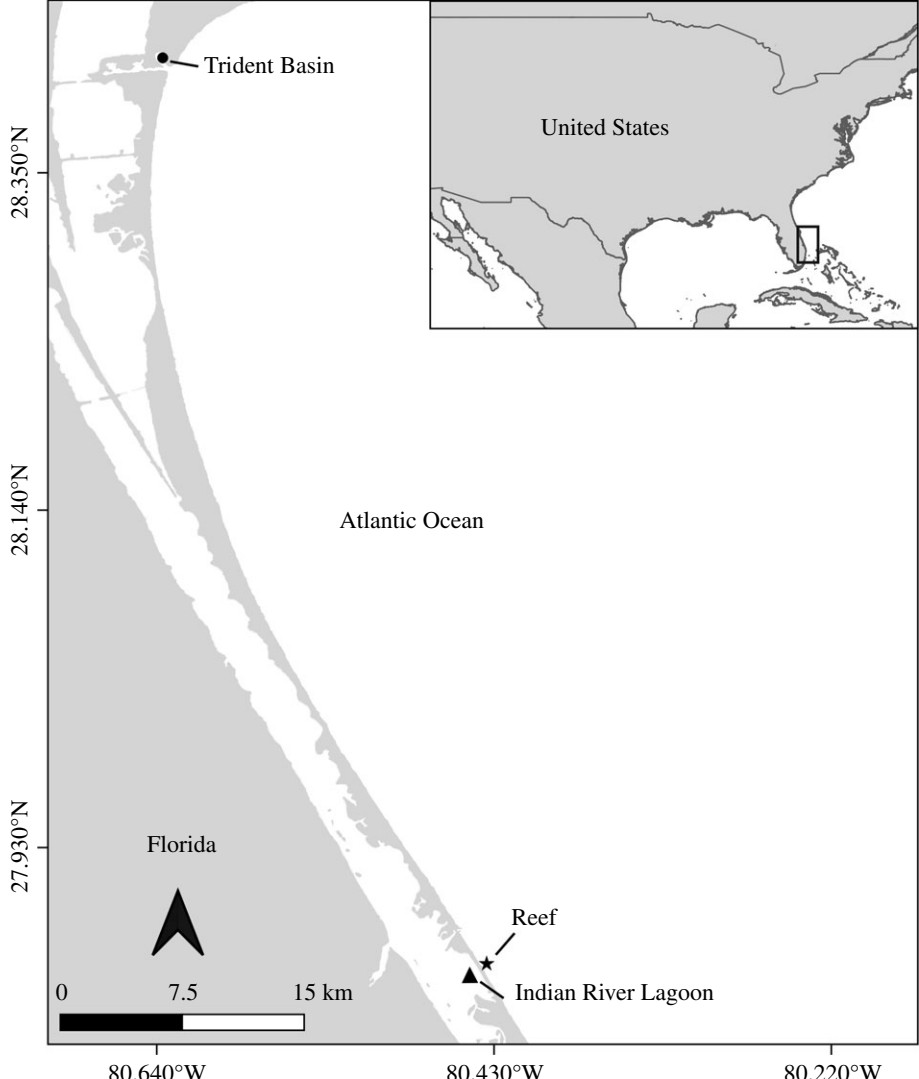

**Figure 1.** Study sites in central Florida where *Ch. mydas* and *Ca. caretta* juveniles are captured and sampled by the University of Central Florida Marine Turtle Research Group. TB is a military port adjacent to Port Canaveral, Florida; the IRL sampling site is located approximately 1–2 miles south of the Sebastian Inlet, Florida. Reef sampling occurred just offshore along the coastal beaches adjacent to the Sebastian Inlet, Florida.

in the IRL [55] and is unknown at the other sites due to low *Ca. caretta* encounter rates at those locations. Here, we leverage these long-term studies to characterize MHC class Iα diversity, molecular evolution and FP associations in Florida *Ch. mydas* and *Ca. caretta* juveniles to understand immunogenetic variation within and across species, and whether it predicts FP susceptibility. We identify novel MHC alleles from *Ch. mydas* and *Ca. caretta* and reconstruct a phylogeny to characterize MHC evolution and diversification. To examine how immunogenetics may affect FP dynamics, we model FP occurrence and smooth tumour texture (which may be associated with disease outcome [34]) in *Ch. mydas* as a function of our MHC and ecological dataset. Our study is the first MHC class Iα characterization in *Ch. mydas* and the first in any juvenile sea turtle, providing a robust foundation to understand immunogenetic shifts in sea turtles as they adapt to disease and other ecological changes.

# 2. Methods

## 2.1. Field methods

The MTRG has sampled the IRL twice a month since 1982 [55], TB twice a year since 1993 [56], and the reef (oceanside, approximately 5 km straight-line from the IRL), opportunistically from 1989 to 2010. We

used blood and skin collected from juvenile *Ch. mydas* and *Ca. caretta* across these sites between 1995 and 2018. Approximately 2 to 5 ml of blood were collected from the dorsal cervical sinus into heparinized blood collection tubes using the antiseptic protocol. Either 20 or 22 gauge, 1 inch needles were used, depending on the size of the turtle. The skin area to be biopsied was first scrubbed with an isopropyl alcohol swab and then a 4 mm sterile biopsy punch was used to sample a portion of the fleshy distal region of the rear flippers. A coagulant powder was used to control bleeding after sampling if needed. Tissues were preserved in 70% ethanol or heparinized tubes, and stored at −20°C for subsequent genetic analysis. The standardized straight carapace length (SCL) of each animal was measured using calipers. External tumours consistent with FP were photographed and scored according to severity [57]. We categorized tumour texture as either rough or smooth based on appearance and used these two categories in subsequent random forest (RF) and relative risk analyses to explore associations between tumour texture and MHC variation. Additionally, individuals that exhibited FP on initial capture that were subsequently recaptured with no external tumours were categorized as regressed ($n = 7$).

## 2.2. Major histocompatibility complex sequencing and bioinformatics

We extracted DNA from blood and skin samples using DNeasy extraction kits (Qiagen, Valencia, CA) and using conventional polymerase chain reaction (PCR), amplified 162 base pair (bp) of exon 2 of classical MHC class I$\alpha$ genes (hereafter, MHC), which encode the $\alpha_1$ subunit and the largest portion of the PBR. We used primers Cc-MHC-I-F (5′-GATGTATGGGTGTGATCTCCGGG-′3) and Cc-MHC-I-R (5′-TTCACTCGATGCAGGTCDNCTCCAGGT-′3) [52]. We performed amplifications in 20 µl reactions using 1 µl DNA template, 14.55 µl of molecular grade water, 2 µl 5x OneTaq Standard Reaction Buffer (New England Biolabs), 2 µl 10 mM dNTPs, 0.8 µl 10% BSA, 0.2 µl 10 mM DMSO, 0.2 µl of each 10 µM primer and 0.25 µl OneTaq DNA polymerase (New England Biolabs). The PCR proceeded as follows: initial denaturation of 95°C for 3 min, 40 cycles of denaturation at 94°C for 30 s, annealing at 52°C for 30 s and extension at 68°C for 30 s, followed by a final extension at 68°C for 1 min. Samples were run on 2% agarose gels to confirm the presence of single amplification products of the correct size (213 bp with sequencing primers).

We ran a second PCR on each 213 bp PCR product to attach Illumina Nextera-style fusion primers to the MHC amplicons. The Stiebens *et al.* [52] primers were optimized for 454 sequencing, thus we redesigned them for the Illumina platform. Our forward and reverse fusion primers consisted of the following (5′ to 3′): Nextera-style Illumina adapters, unique 8 bp indices, a 10 bp primer pad, 2 bp linker and the MHC sequencing primers described above. PCRs were performed with unique combinations of forward and reverse fusion primers for MHC amplicons from each individual in a 25 µl reaction, using 4 µl of the PCR product from the first amplification, 6.875 µl molecular grade water, 2.5 µl 5x OneTaq Standard Reaction Buffer (New England Biolabs), 0.5 µl 10 mM dNTPs, 1 µl 10 mM forward fusion primer, 10 µl 1 mM reverse fusion primer and 0.125 µl OneTaq DNA polymerase (New England Biolabs). The PCR was run at an initial denaturation of 95°C for 3 min, 15 cycles of denaturation at 94°C for 30 s, annealing at 52°C for 30 s and extension 68°C for 30 s, followed by a final extension at 68°C for 1 min. Samples were visualized on 2% agarose gels to confirm product size (307 bp including fusion primers) and pooled into three groups based on band intensity (strong, medium and weak). Each pool was cleaned and DNA concentration was quantified, and a final pool was created. We used 1.0 X SpeedBeads (Thermo Fisher Scientific, Waltham, MA) to clean the amplicon pools, which were then quantified with a KAPA qPCR kit following manufacturer's instructions (Roche Sequencing Systems, Pleasanton, CA). The final library pools, each containing roughly equimolar quantities of MHC amplicon from each individual, were sequenced across three Illumina MiSeq 2 × 250 runs, twice at the Oklahoma Medical Research Foundation Clinical Genomics Center and once at the University of Central Florida Genomics and Bioinformatics Cluster. Five individuals were separately amplified and sequenced on the three runs for repeatability to identify possible amplification and sequencing biases. Demultiplexed and adapter-trimmed reads were returned from the sequencing machine and used in downstream analyses.

Paired-end MHC reads were merged using PEAR v. 0.9.11 [58] with default parameters, including a Phred trim quality score of 33. Merged reads were then clustered into MHC alleles using AmpliSAS with the developer's parameter recommendations for Illumina data [59]. We compared maximum allele limits ranging from 1 to 12 alleles. Our minimum amplicon depth was 160 reads and minimum amplicon frequency was 10% for clustering and 3% for filtering. We analysed resulting alleles using Geneious v. 9 [60], removing any allele that was not 162 bp (our fragment size after removing primers), not

recovered in at least two individuals, contained stop codons, or did not have significant BLAST hits with any MHC class I$\alpha$ sequence.

## 2.3. Major histocompatibility complex genetic variation

MHC class I$\alpha$ variability was evaluated separately in alleles recovered from *Ch. mydas* ($n$ = 98 alleles) and from *Ca. caretta* ($n$ = 32 alleles) in our study. We calculated the average number of alleles per individual in each species and used the nuc.div() and seg.sites() functions from the pegas v. 0.12 [61] and ape v. 5.3 [62] packages in R to calculate nucleotide diversity and number of segregating sites in alleles per species, respectively. We were unable to calculate metrics of heterozygosity because our multi-locus alleles cannot be assigned to specific loci.

## 2.4. Phylogenetic and network analyses

We used Clustal Omega [63] in Mega v. 7.0.26 [64] to align the sequences. We used MHC sequences from *Gallus gallus* (KF032390.1) and *Tymphanuchus cupido* (KF466478.1) as outgroups and included the eight unique (after removing primer sequence) MHC alleles that were not found in our study and were previously sequenced from a *Ca. caretta* nesting population in Cape Verde, Africa (GenBank IDs: KF021627, KF021629, KF021631–KF021637, KF021639–KF021646, KF021649–KF021651, KF021661, KF021663, and KF021665) [52]. We used PartitionFinder v. 2.1.1 [65] to find the model of evolution that best fit the alignment and reconstructed a Bayesian phylogeny using MrBayes v. 3.2.6 [66] run twice for $1 \times 10^7$ generations each, with the first 100 000 iterations discarded as burn-in. Results were visualized in Tracer v. 1.7 [67] to confirm MCMC chain convergence and ensure adequate posterior distribution sampling. We reconstructed a maximum-likelihood phylogeny in RAxML v. 8.2.8 [68] and visualized both phylogenies using the ggtree package [69]. We conducted a haplotype network analysis in PopArt v. 1.7 using the TCS algorithm to visualize allelic frequencies and distributions [70].

## 2.5. Major histocompatibility complex supertyping

We used discriminant analysis of principal components (DAPC) to identify MHC supertypes based on the physiochemical properties of the amino acid residues in our MHC alignment [71–73]. Using a two-step cross-validation and optimization procedure, we iteratively evaluated and cross-validated the number of principal components to retain at $k$ clusters 3 to 11 using the R package adegenet v. 2.1.2 [74] (electronic supplementary material, methods S1, figures S9–S18).

## 2.6. Random forest modelling

RF ensemble learning models provide a non-parametric analytical approach to incorporate many explanatory variables and their associations to predict a response variable [75]. For polygenic loci such as MHC, we opted for RF modelling to explore how many explanatory variables of interest, including numerous MHC alleles, might predict disease state in sea turtles. We implemented RF models to predict (i) FP occurrence and (ii) tumour texture, each as a function of the following variables: the presence of each MHC allele, the presence of each MHC supertype, total number of MHC alleles per individual, season (spring, summer, autumn or winter), year, capture location and SCL. Because we had few FP-positive *Ca. caretta* ($n$ = 4 out of 88), we conducted these analyses on *Ch. mydas* only. We also corrected imbalance in the response variable of both models by subsampling. For the FP occurrence prediction, we considered any turtle encountered with FP at any point in time to be FP positive. For the tumour texture prediction, we included FP positive *Ch. mydas*, coding individuals as having either rough or smooth FP tumours based on tumour description at time of capture [34]. We did not include the seven individuals with true tumour regression in the RF analyses due to small sample size of this category ($n$ = 7). We included data (e.g. SCL, season, year) from the first capture for recaptured turtles, or if applicable, from the capture when FP was observed. Additionally, only individuals that had SCL measurements were included in the analyses ($n$ = 268 for the FP occurrence model, $n$ = 99 for the tumour texture model). RF analyses were implemented in RStudio using the packages randomForest v. 4.6.14 [76] and caret v. 6.0.86 [77]. We used optimized values for the *mtry* and *ntree* parameters, which dictate the number of variables available to split nodes in each RF decision tree and the total number of trees, respectively. For the final models predicting FP occurrence and tumour texture, the data were randomly separated into 30% test and 70% training datasets. For

FP occurrence, the model was built on the training data with $mtry = 17$ and $ntree = 10\,000$. For tumour texture, the final model was built with $mtry = 50$ and $ntree = 10\,000$. For both models, a confusion matrix was built and the prediction accuracy of each model was assessed, as well as the relative importance of the predictor variables based on two metrics: mean decrease in accuracy and mean decrease in Gini impurity (see electronic supplementary material, methods S2).

## 2.7. Relative risk

We conducted risk ratio analyses to explore the MHC-based relative risk of individuals (i) developing FP and (ii) smooth-textured tumours. We included only *Ch. mydas* due to insufficient sampling of FP-positive *Ca. caretta*, and only retained alleles that were found in 10 or more individuals due to low statistical power for small sample sizes [78]. We also assessed relative risk for FP and for smooth tumour texture based on the three MHC supertypes. The seven individuals with true tumour regression were excluded from the tumour texture risk ratio analyses due to small sample size of this category ($n = 7$). Analyses were conducted in RStudio with the package epiR v. 1.0.14 [79] using the epi2by2 function with cohort method and 95% confidence intervals. For any allele for which there was a zero cell (i.e. one of the categories had a count of zero), a Haldane-Anscombe correction was applied [80,81]. Bonferroni correction for multiple comparisons was applied to keep the family-wise error rate at 5% [82].

## 2.8. Selection analyses

To identify positive selection within the MHC alignment, we implemented HyPhy on the DataMonkey server [83,84], inputting our translated MHC alignment (53 codon alignment of 124 alleles) and Bayesian phylogeny. Model selection was performed within HyPhy to determine which model of DNA evolution best fit the data, and single breakpoint analysis was used to identify intragenic recombination, which can bias selection measurements [83]. Upon detecting significant evidence of recombination, we partitioned the alignment at the recombination breakpoint. For each partition, we identified a model of evolution with PartitionFinder v. 2.1.1 [65] and reconstructed a Bayesian phylogeny using MrBayes v. 3.2.6 [66] with the same methods used for the phylogenetic analysis described previously. We then conducted separate tests of selection on the two partitions and phylogenies, including the following site-specific methods: FEL (Fixed Effects Likelihood), SLAC (Single Likelihood Ancestry Counting) [85], FUBAR (Fast Unconstrained Bayesian AppRoximation) [86] and MEME (Mixed Effects Model of Episodic Diversifying Selection) [87]. We also implemented aBSREL and BUSTED, which are branch and gene-wide tests, respectively, to test for signatures of selection on branches of the MHC phylogeny. aBSREL and BUSTED provide statistical evidence for positive selection on branches of the phylogeny but do not identify individual sites. Significance was inferred at alpha less than or equal to 0.05 or posterior probability greater than or equal to 0.95, as appropriate for the method used.

# 3. Results

## 3.1. Major histocompatibility complex sequencing and bioinformatics

We generated robust MHC genotype data for 356 of the 443 juvenile sea turtles we evaluated (268 *Ch. mydas* and 88 *Ca. caretta*) and recovered a maximum of seven MHC alleles per individual (when allowing detection of up to 12 alleles per turtle). Allele counts per turtle ranged from one to seven alleles, with a mean of $3.6 \pm 1.4$ alleles (electronic supplementary material, figure S1). In total, we recovered 116 unique MHC alleles (corresponding to 99 unique translated amino acid alleles), of which 15 were identical to previously identified alleles in Cape Verde *Ca. caretta* [52]. Among the samples amplified and sequenced independently, four returned identical results while one differed slightly between sequencing runs, with five alleles identified in run 1 and four of those same five alleles identified in run 2, when total sample read depth was lower (electronic supplementary material, table S1). Eighty-four alleles occurred only in *Ch. mydas* and six occurred only in Florida-sampled *Ca. caretta*. Eleven alleles were shared by *Ch. mydas* and *Ca. caretta* from Florida, 12 alleles were shared by Florida and Cape Verde *Ca. caretta*, and three alleles were shared by *Ch. mydas*, Florida *Ca. caretta* and Cape Verde *Ca. caretta*. Among MHC-genotyped *Ch. mydas*, 106/268 (39.6%) individuals in our dataset had FP at the time they were sampled. FP occurrence was much lower in *Ca. caretta*, with only 4/88 (4.5%) of individuals manifesting FP at the time of sampling.

**Table 1.** Estimates of MHC genetic diversity for alleles recovered in *Ch. mydas* and *Ca. caretta*, including turtle sample sizes (N), allele sample size (n), average number of alleles observed per individual, segregating sites over 162 bp and nucleotide diversity.

| species | turtles sampled (N) | alleles sampled (n) | average number of alleles observed (s.d.) | segregating sites | nucleotide diversity |
|---|---|---|---|---|---|
| *Ch. mydas* | 268 | 98 | 3.5 ± 1.4 | 106 (65.43%) | 0.201021837 |
| *Ca. caretta* | 88 | 32 | 3.9 ± 1.5 | 95 (58.64%) | 0.2347795 |

## 3.2. Major histocompatibility complex genetic variation

Among MHC alleles recovered from *Ch. mydas*, there were 106 segregating sites out of the 162 total sequenced nucleotides (65.4%) and nucleotide diversity was 0.20. Similarly, among MHC alleles recovered from *Ca. caretta* there were 95 segregating sites out of 162 (58.6%) and nucleotide diversity was 0.23 (table 1).

## 3.3. Phylogenetic reconstruction, major histocompatibility complex supertyping and haplotype network analysis

We identified three well-supported clades of MHC alleles (figure 2; electronic supplementary material, figure S2). DAPC clustered the 124 alleles into three distinct supertypes (electronic supplementary material figures S10A, S10B), which correspond precisely to these three clades. The supertype A clade had not previously been identified, whereas alleles from supertype B and C clades were recovered in the previous study of *Ca. caretta* MHC [52]. Alleles did not cluster by species (colours), by FP status (asterisks), or by geographic location (Florida versus Cape Verde) (figure 2). A majority of the 116 alleles recovered in this study were found in only a few individuals, but some occurred at high frequency, particularly within supertype B (figure 3). All alleles that occurred in both species were primarily recovered in either *Ch. mydas* or *Ca. caretta*, and were never common in both species (figure 3).

## 3.4. Random forest modelling

The RF model predicting *Ch. mydas* FP occurrence had an out-of-bag error rate of 37.04% and prediction accuracy of 67.38%. Variables that had the highest relative importance in predicting FP were location, SCL, year, and season. Individual MHC alleles showed lower relative importance, although allele Chmy49 and allele count had the highest relative importance of any MHC variable based on mean decrease accuracy and mean decrease Gini impurity, respectively (figure 4). The RF model predicting *Ch. mydas* FP tumour texture had an out-of-bag error rate of 26.67% and prediction accuracy of 66.67%. The predictor variables with the highest relative importance based on mean decrease accuracy and mean decrease Gini impurity included year, SCL, allele count, and MHC alleles Chmy04 and Chmy80 (figure 5).

## 3.5. Relative risk

*Chelonia mydas* with allele Chmy13 had a significantly elevated risk ratio for having FP (RR = 1.74; 95% CI = 1.13–2.68; electronic supplementary material, figure S5), although this was not significant after Bonferroni correction (RR = 1.74; 99.8% CI = 0.885–3.43). Among FP positive *Ch. mydas*, individuals with alleles Chmy04 had a significantly elevated risk ratio for having tumours with smooth texture (Chmy04: RR = 2.28, 95% CI = 1.08–4.81; electronic supplementary material, figure S6), but this was not significant after Bonferroni correction (RR = 2.28; 99.9% CI = 0.851–6.09).

*Chelonia mydas* with MHC supertype B had a significantly decreased risk for having FP (RR = 0.731, 95% CI = 0.544–0.982; electronic supplementary material, figure S7), but this was not significant after Bonferroni correction (RR = 0.731, 98.3% CI 0.510–1.05). There were no significant associations between smooth tumour texture and MHC supertype (electronic supplementary material, figure S8).

## 3.6. Selection analyses

Across the 124 alleles, seven codons in the 53 codon alignment evolved under positive selection based on at least one site-based method (table 2; electronic supplementary material, figure S3) and 4/4 site-based

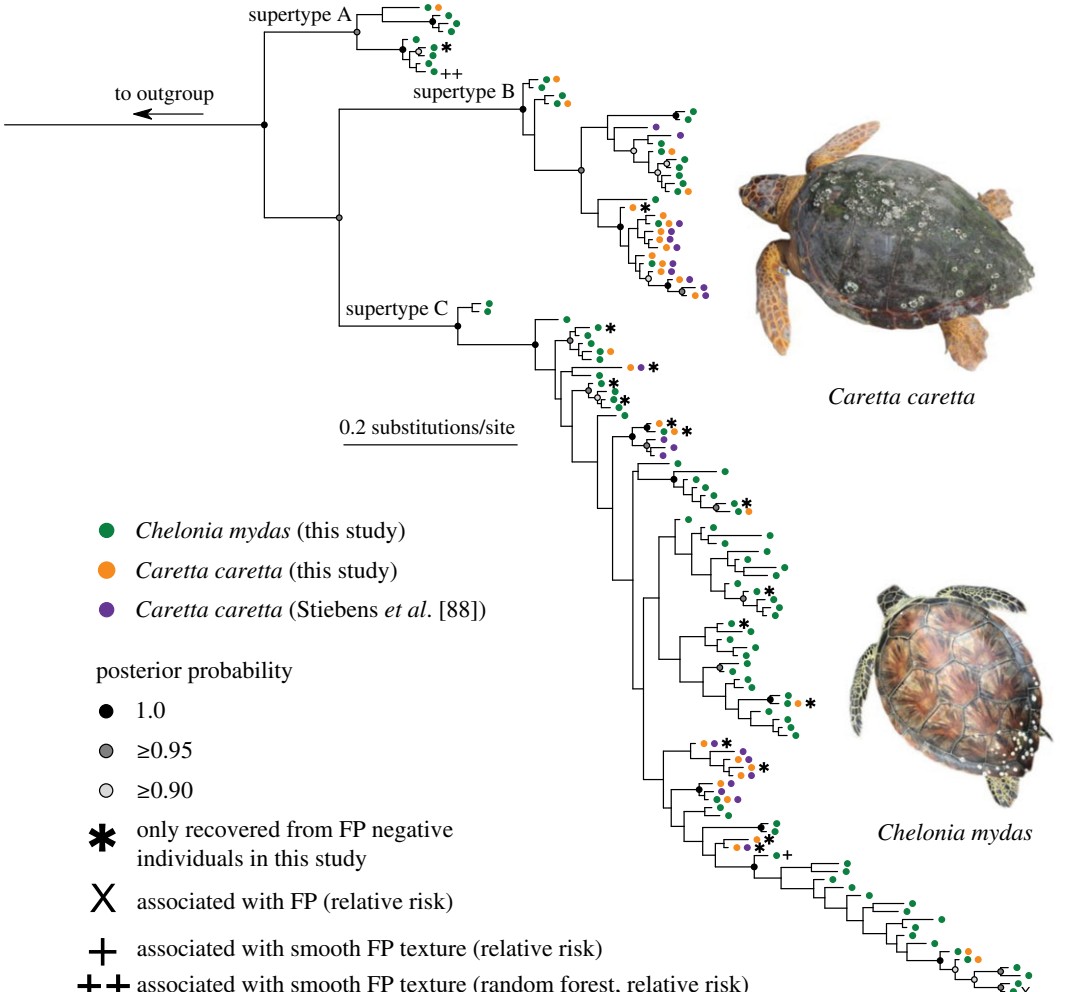

**Figure 2.** Bayesian phylogeny of 124 alleles of exon 2 (162 bp) of the Iα MHC in *Ch. mydas* and *Ca. caretta* (116 of which were recovered from turtles sampled in this study from the central Florida Atlantic coast). Posterior probabilities greater than 0.90 are shown and supertype group membership is denoted on interior branch. Asterisk denotes alleles recovered in FP negative turtles. 'X' denotes alleles significantly with FP in relative risk analysis. Single cross symbol denotes alleles associated with smooth tumour texture per relative risk analysis; double cross symbol denotes alleles associated with smooth tumour texture per random forest modelling and relative risk analysis.

methods identified codon 51 evolving under positive selection. BUSTED detected episodic diversifying selection in at least one site on at least one branch of the phylogeny, whereas aBSREL did not detect any evidence of episodic diversifying selection in the phylogeny. Across our alleles, six unique amino acids occur at codon 51: aspartic acid, glutamic acid, glutamine, arginine, tryptophan and tyrosine. Some clades in our phylogeny include a wide variety of these six amino acids at position 51, whereas other clades have only one or two amino acids (electronic supplementary material, figure S4). The allele not significantly associated with FP occurrence in the relative risk analyses, Chmy13, has a tryptophan at codon 51, which is amphipathic. The alleles weakly associated with smooth tumour texture from the RF analyses have arginine (Chmy04), and tryptophan (Chmy80) at codon 51, which are charged and amphipathic, respectively.

## 4. Discussion

Our study provides the first characterization of MHC immunogenetic diversity in any life stage of *Ch. mydas* and the first in juvenile *Ca. caretta*. We recovered high MHC class Iα polymorphism in both species based on sequence diversity and number of recovered alleles. Low immunogenetic diversity following population bottlenecks has been documented in amphibians [89], birds [90] and non-avian reptiles [53]; however, despite historical reductions in *Ch. mydas* population sizes [31] and threats to

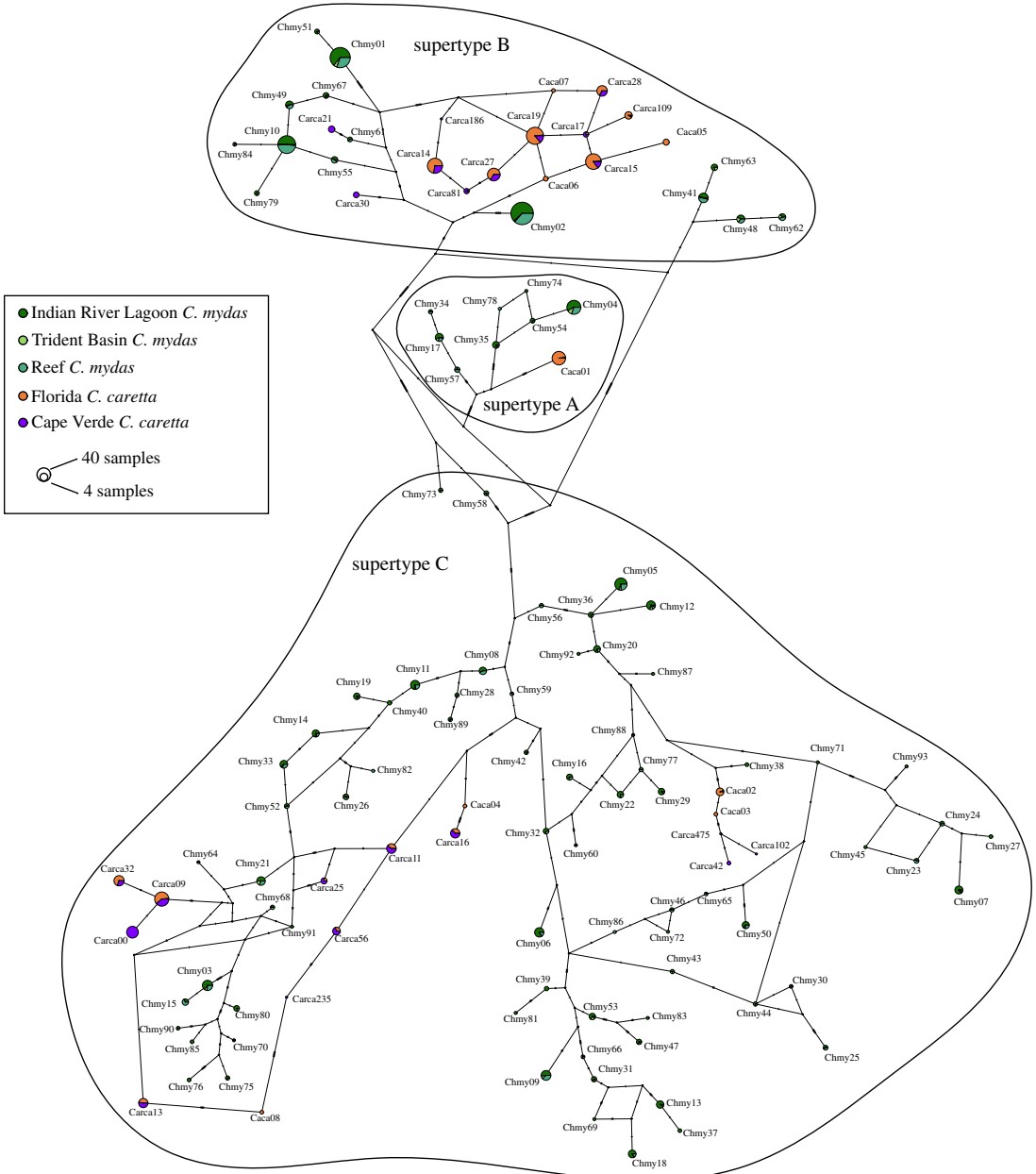

**Figure 3.** Haplotype network of 124 class I exon 2α (162 bp) MHC alleles recovered in *Ch. mydas* and *Ca. caretta* juveniles. Circle size in proportional to overall allele frequency, and alleles are coloured proportionally according to species and field study location (Florida or Cape Verde for *Ca. caretta*; IRL, Trident or Reef for *Ch. mydas*). Supertypes are denoted with black outlines. Allele frequencies for Cape Verde *Ca. caretta* from [88] and C. Eizaguirre (personal communication).

*Ca. caretta* [91], we found no evidence of low immunogenetic diversity in sea turtles. We found no significant associations between MHC alleles and FP tumour development or tumour texture. Because standing immunogenetic variation is critical for the survival of disease-threatened organisms, as it allows ongoing adaptation to changing pathogen pressures [16], we were surprised to find only minimal MHC-disease associations in our study. However, the high number of recovered alleles made sample sizes small for most individual allele-disease associations, and we hope future studies will continue to characterize MHC variation in the context of FP as we suspect larger allele sampling could reveal stronger relationships. Alternatively, MHC may not function in sea turtles as documented in other taxa, and characterizing non-MHC immune genes and immune function are important priorities given the threat posed by FP, especially in the severely affected *Ch. mydas* [26,34] and in juveniles newly recruited to neritic waters [92]. Recent functional immune studies demonstrate reduced natural killer cell response in *Ch. mydas* with FP [39] and reduced phagocytosis in individuals

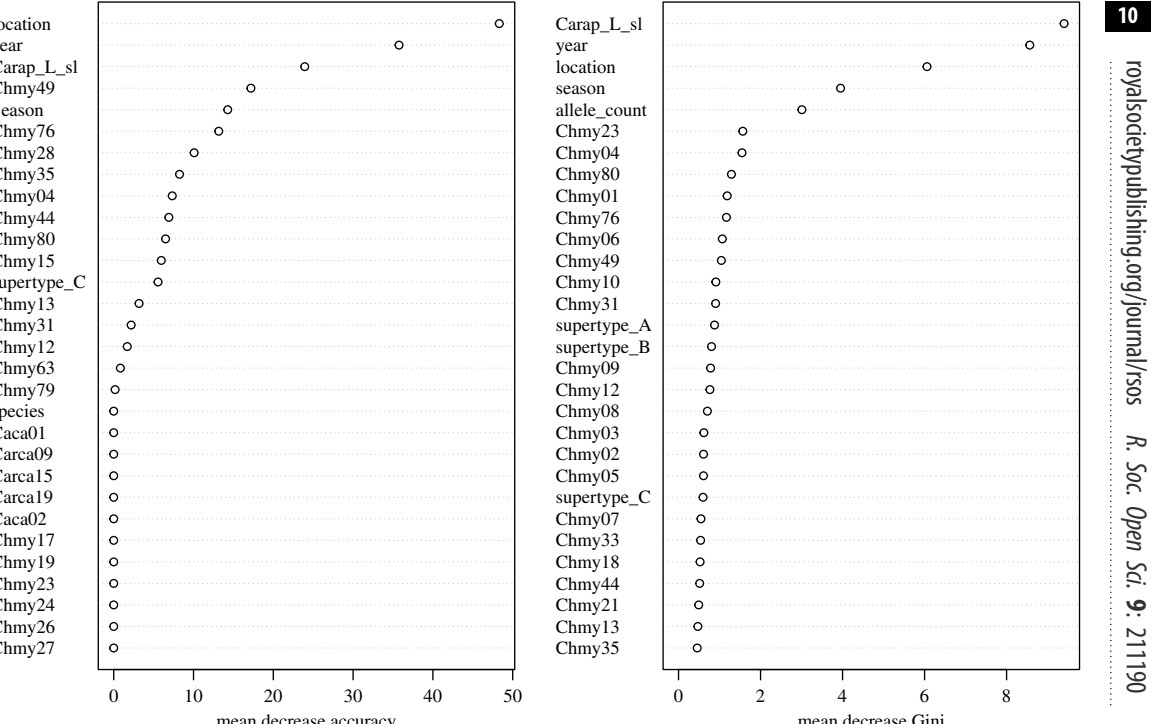

**Figure 4.** Mean decrease in accuracy (*a*) and Gini impurity (*b*) for random forest model relating FP occurrence in *Ch. mydas* from the central Florida Atlantic coast to MHC genetic diversity predictors.

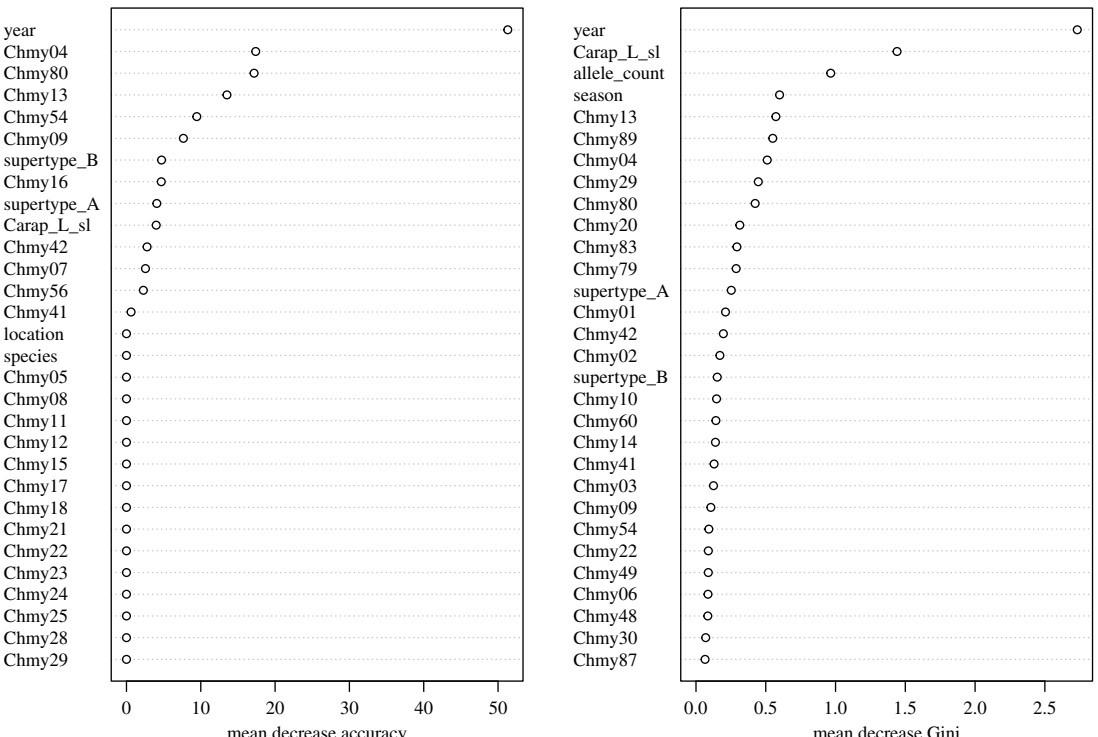

**Figure 5.** Mean decrease in accuracy (*a*) and Gini impurity (*b*) for RF model relating tumour texture in *Ch. mydas* from the central Florida Atlantic coast to MHC genetic diversity predictors.

in degraded habitat, regardless of FP status [42]. Additionally, several gene expression studies in *Ch. mydas* indicate that FP is associated with changes in immune gene expression, including upregulated immune checkpoint molecules in tumours [43], heightened T-lymphocyte activity in early-stage tumours and distinct expression patterns in internal versus external tumours [44],

**Table 2.** HyPhy branch- and site-specific tests of positive selection in the codon alignment of exon 2 (PBR) of MHC class Iα in *Ch. mydas* and *Ca. caretta*. Evidence of positive selection at each codon position is denoted with an X, with method and statistical criteria specified.

| method | statistical cut off | codon position | | | | | | |
| --- | --- | --- | --- | --- | --- | --- | --- | --- |
| | | 2 | 3 | 10 | 18 | 36 | 42 | 51 |
| aBSREL | $p \leq 0.05$ | no evidence of episodic diversifying selection across phylogeny | | | | | | |
| FEL | $p \leq 0.05$ | X | | | | | X | X |
| FUBAR | posterior probability $\geq 0.95$ | | | X | X | | | X |
| MEME | $p \leq 0.05$ | X | X | X | | X | | X |
| SLAC | $p \leq 0.05$ | | | | | | | X |
| BUSTED | $p \leq 0.05$ | gene-wide episodic diversifying selection on at least one branch | | | | | | |

upregulated tumour inhibition transcripts in FP-afflicted turtles [45] and alteration of signalling pathways in FP tumours that have also been implicated in human cancers [46]. Together, these studies demonstrate the importance of understanding the immune system's role in FP, and we advocate for the continued evaluation of sea turtle immunogenomic repertoires using gene expression analysis, functional immune assays and immune gene characterization.

Our RF modelling had low predictive power overall (67.38% for FP development and 66.67% for smooth tumour texture). MHC allele Chmy13 and MHC supertype B were associated with FP risk in the relative risk analyses, but neither association was significant after Bonferroni correction. For tumour texture, MHC allele Chmy04 was the second highest ranked variable for decrease in mean accuracy in the RF model and was associated with an increased risk of smooth tumour texture in the relative risk analysis, although this was not significant after Bonferroni correction. Further sampling of Chmy04 and the other weakly disease-linked MHC variables identified here will be important for confirming any associations with FP development and tumour texture. In a recent study of host and viral gene expression in *Ch. mydas*, Blackburn *et al.* [43] found that two tumours with the highest levels of ChHV5 transcripts (suggesting tumour growth) had smooth texture and altered pigmentation. By contrast, smooth tumour texture was anecdotally linked to tumour regression in juvenile *Ch. mydas* sampled in the IRL [34]. While we cannot conclusively link tumour texture to either FP progression or regression, we suggest that tumour appearance and its relation to host immune function, tumour gene expression, and disease outcome is an area of exploration for future studies. Moreover, immunogenetic and functional immune studies in sea turtles with true regression (i.e. shrinking or disappearance of tumours from initial capture to subsequent recapture) may also provide insight into the mechanisms behind the variable outcomes of FP. Given the current modest support linking MHC and FP, and the complexity of the vertebrate immune system, immunogenetic associations with FP may largely be driven by other immune loci and/or immunogenetic–environmental interactions. Based on previous transcriptome-wide studies in sea turtles [43–46,93], gene expression studies could be especially illuminating in evaluating the role of MHC expression relative to tumour development, where MHC expression may be a better predictor of FP status than the presence or absence of specific alleles. FP is rarely documented in species other than *Ch. mydas*, making species-level comparisons of disease incidence difficult. We suggest that sampling FP-positive *Ca. caretta* should be a priority in sea turtle disease studies to provide insight into host differences that may translate into different rates of juvenile FP documented between species at our field sites. Despite the limitations of our disease analyses, our study provides a valuable survey of MHC diversity in two sea turtle species, and a first step in understanding potential relationships between immunogenetic variation and disease occurrence in sea turtles.

We detected high MHC class Iα polymorphism in both juvenile *Ch. mydas* and *Ca. caretta*, bringing the number of unique alleles identified in sea turtles to 124 [52,88]. In both species, the percentage of segregating sites was over 50%. However, this polymorphism is potentially overestimated, as the alleles we recovered come from multiple unidentified loci. By comparison, studies of Testudines that were able to assign MHC class I alleles to loci found that across 18 populations of Gopher tortoise (*Gopherus polyphemus*) at MHC class I exon 3, there were 13% segregating sites and nucleotide

diversity was 0.03 [53]; furthermore, in a population of Diamondback terrapins (*Malaclemys terrapin*), the MHC class I locus was fixed for a single allele [50]. By contrast, our conservative criteria for retaining alleles means we are probably underestimating total alleles, and additional sea turtle MHC studies will almost certainly reveal further MHC class Iα diversity. Also, given documented MHC copy number variation among individuals in other taxa [90,94], including in reptiles (e.g. birds) with up to 33 loci [95], further sea turtle MHC sampling could find that the maximum number of alleles per individual may exceed seven. However, the exploration of MHC genomic structure in sea turtles using new genomic resources such as well-annotated assemblies of the *Ch. mydas* genome (rCheMyd1; RefSeq assembly accession GCF_015237465.1) and implementation of bioinformatics approaches [96] will be critical for resolving the extent of MHC copy number variation in sea turtles.

We found both sequence and allelic diversity patterns that are generally consistent with MHC class Iα evolving adaptively via balancing selection to maintain polymorphism [97]. In studies of MHC, patterns consistent with balancing selection are usually interpreted in the context of pathogen-mediated balancing selection, where pathogen pressure selects for high levels of MHC polymorphism via several non-mutually exclusive selective processes, including heterozygote advantage, frequency-dependent selection (or rare allele advantage) and fluctuating selection [12,16], although distinguishing between the different mechanisms of pathogen-mediated balancing selection is very difficult [98].

At the sequence level, we detected some evidence of positive selection at seven MHC codons, and all four site-specific tests identified codon 51 as evolving under positive selection. An increased rate of non-synonymous relative to synonymous substitutions at MHC peptide-binding residues is consistent with pathogen-mediated selection [99]. While codon 51 is not homologous to a known peptide-binding residue when aligned to the human reference, our results suggest it is the most important target of adaptive evolution in sea turtle MHC class Iα and may play an important role in reptilian MHC pathogen-binding. Across our sampled alleles, six different amino acids with distinct-binding abilities occurred at codon 51, which could be indicative of variable pathogen-binding ability at this residue across MHC alleles, perhaps driven by fluctuating selection or changing pathogen pressures. However, unless stronger relationships between FP and specific amino acid changes in MHC alleles are found, we cannot robustly link these signatures of positive molecular selection or amino acid variation at codon 51 to pathogen pressure. Numerous studies of MHC class II evolution evaluate specific codons under selection and find consistent amino acid changes and associations with disease [17,20,100,101], but to our knowledge, only one other study has established similar codon-disease associations in reptilian class I loci [49]. Thus, we encourage similar studies across reptilian taxa to provide a better framework for drivers of MHC class I adaptive evolution and to identify common targets of pathogen-mediated selection.

Previous studies suggest that individuals with intermediate numbers of MHC alleles may have greater fitness compared with those with maximized heterozygosity [102,103], where high MHC diversity may be selectively advantageous for pathogen binding but is disadvantageous because it leads to greater T-cell repertoire depletion to prevent autoimmunity [98,104]. While the average number of alleles we recovered per turtle is generally consistent with this pattern of intermediate allele numbers, we did not find statistical support for an association between FP and number of alleles in sea turtles. Further, we caution that PCR amplification biases often affect estimates of MHC allele count [105,106], limiting confidence in estimates of allele counts per individual.

The total number of alleles we recovered from both species is high, which is consistent with expectations that balancing selection will maintain high levels of allelic polymorphism. However, the aggregations of *Ch. mydas* and *Ca. caretta* juveniles that recruit to neritic foraging waters after their early oceanic stage [34,55] represent mixed genetic stocks from multiple source rookeries [107] which may be driving the sheer number of recovered MHC alleles in our study. Only a few dozen mtDNA haplotypes have been recovered from the entire northwest Atlantic region, most differing from each other by a single nucleotide [108], suggesting that the high MHC polymorphism we recovered is not necessarily caused by sampling particularly diverse genetic lineages. However, we caution against over-interpretation given that mitochondrial and MHC loci are subject to different demographic processes; evaluation of the molecular evolution of numerous nuclear loci and assignment of individual MHC alleles to loci are needed before we can robustly quantify the genetic diversity present within our study area or describe the evolutionary forces that shaped it. Finally, gene flow between source populations with different levels of local MHC polymorphism may also be inflating MHC diversity estimates and obscuring potential associations with FP.

The pattern of shared MHC alleles we found in *Ch. mydas* and *Ca. caretta* is also consistent with expectations under balancing selection. The three clades of alleles we recovered did not cluster by

species; instead 14 identical alleles occurred in both species (including alleles sampled from Cape Verde *Ca. caretta* [52]), and many *Ch. mydas* and *Ca. caretta* alleles shared closer evolutionary histories with each other than with other alleles from the same species (figure 2). We note with interest that this allele sharing spans across the subfamilies Carettinae and Cheloniinae and that shared MHC polymorphism between genera or higher taxonomic ranks is only rarely documented [109–111]. Shared alleles are potentially explained by trans-species polymorphism [112], wherein standing ancestral MHC polymorphism is maintained through speciation due to incomplete lineage sorting [113] and is subsequently preserved by balancing selection in the descendant species [114]. Alternatively, shared alleles among species may also originate through hybridization; in the case of MHC, novel allele advantage is thought to promote adaptive introgression [115,116]. While *Ch. mydas* and *Ca. caretta* speciated approximately 63 million years ago [117], hybridization with viable offspring occurs between Cheloniidae species [118,119]. Furthermore, *Ch. mydas* × *Ca. caretta* hybrids have been documented [120], including a first generation hybrid in our TB field site [121]. We suggest that introgression between *Ch. mydas* and *Ca. caretta* could explain at least part of this pattern. However, shared alleles between species may also arise from convergent evolution or from nucleotide changes that result in reversions to the ancestral state, and distinguishing the mechanisms by which shared alleles arise in related taxa is difficult in practice [113]. Finally, we acknowledge that sea turtle MHC polymorphism is probably shaped by multiple factors and selective pressures working simultaneously, and that disentangling them will require focused sampling across time and populations, as well as sampling neutral marker data for comparison with MHC evolution. We note that balancing selection also encompasses other selective pressures such as assortative mating and maternal–fetal interactions that may work to maintain MHC polymorphism [97]. While testing for these phenomena is beyond the scope of this study, they may be valid avenues of research given the limited evidence we found for a relationship between sea turtle MHC and FP.

Reptile immune systems are relatively understudied by comparison with birds and mammals [51]. Our study underscores this need for more studies of reptile immunogenetics: with such high numbers of MHC alleles, we lack the sampling depth at each allele to robustly explain FP status. Furthermore, the innate immune system is postulated to be more robust than the acquired immune system in reptiles due to life history and ectothermy [122,123], and we suggest that investigations into non-MHC immune genes and genes associated with the innate immune system (e.g. natural antibodies [124]) may be fruitful next steps for understanding FP in sea turtles, and more broadly in understanding how immunogenetics moderates disease in reptiles.

Coastal habitats are an important incubator of genetic diversity in sea turtles [107], and our study suggests they may also be a haven of immunogenetic diversity, although similar studies from other regions are necessary to determine whether MHC diversity is equally high in other sea turtle aggregations and populations. Juvenile sea turtles face numerous threats, including nutrient pollution in foraging grounds [125], historical population contractions from overexploitation [31], and disease [36]. Disease is a particular problem in coastal sea turtle foraging habitat, where pathogen-naive juveniles are proposed to be initially exposed to the ChHV5 virus after recruitment from oceanic waters [92] and where pollution may exacerbate disease [126]. Standing levels of immunogenetic variation at the coastal juvenile life stage may very well be a deciding factor in survival and future fitness. We advise that monitoring the immunogenetic diversity of both source populations and juvenile aggregations is important given that delayed sexual maturity and long generation intervals in sea turtles may limit their response to environmental pressures (e.g. ChHV5), warranting conservation measures at multiple life stages [127]. Future studies characterizing immunogenetic diversity in other life stages and species of sea turtles should be conducted to improve our understanding of sea turtle immune systems and that of reptiles more broadly. Reptilian immune systems are poorly studied relative to other vertebrate taxa [51], and although expression changes in immune-related genes have been consistently linked to FP tumour development and sea turtle outcome [43–46], the roles of the innate and acquired immune systems in fighting disease remain open questions in reptile biology [128]. Our study demonstrates the extent of MHC polymorphism that has evolved in sea turtles despite an apparently weak association with the FP disease and illustrates that the study of reptilian MHC can broaden our understanding of how the vertebrate immune system evolves and functions.

Ethics. All samples were collected under National Marine Fisheries Service permit no. 19508 and Florida Marine Turtle Permits 231 and 225 (and all associated predecessors), as well as approved institutional animal care and use committee protocols.

Data accessibility. Data and relevant code for this study are stored on GitHub: https://github.com/katherinermartin/
Martin_et_al_RSOS_2021 and archived within the Zenodo repository: https://doi.org/10.5281/zenodo.5703775
[129]. The sequence data are available in GenBank, accession numbers OK135205–OK135305. The data are provided
in the electronic supplementary material [130].

Authors' contributions. K.R.M.: data curation, formal analysis, investigation, project administration, writing—original draft,
writing—review & editing; K.L.M.: conceptualization, funding acquisition, project administration, resources,
supervision, writing—review & editing; A.E.S.: conceptualization, funding acquisition, methodology, project
administration, resources, supervision, writing—review & editing

All authors gave final approval for publication and agreed to be held accountable for the work performed therein.

Competing interests. We declare we have no competing interests.

Funding. This work was supported by the Florida Sea Turtle License Plate Program which is funded by the proceeds of
the Florida Sea Turtle License Plate (award no. 18-033R, 2018–19). Article processing charges were provided in part by
the UCF College of Graduate Studies Open Access Publishing Fund.

Acknowledgements. We are grateful to the many members of the MTRG over the last 40 years for their dedication to
collecting and cataloguing samples; M. Lawrance, E. Sutton, and A. Trujillo for their help in primer design, DNA
extraction, and generating pilot data; R. Fitak for assistance with statistical analysis; and V. Urgiles for help with
figures and maps. We also thank members of the Savage lab, MTRG, and several anonymous reviewers for their
feedback on earlier versions of this manuscript.

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
