## [Peer Review File · Royal Society Open Science]

Review History

RSOS-211190.R0 (Original submission)

Review form: Reviewer 1

Is the manuscript scientifically sound in its present form?

Yes

Are the interpretations and conclusions justified by the results?

Yes

Is the language acceptable?

Yes

Do you have any ethical concerns with this paper?

No

Have you any concerns about statistical analyses in this paper?

No

Recommendation?

Accept with minor revision (please list in comments)

Comments to the Author(s)

Thank you for the opportunity to review "Adaptive evolution of MHC class I immune genes and disease associations in coastal juvenile sea turtles." In addition to my review of the current draft of the manuscript, I reviewed the comments of the two previous reviewers and the responses and edits provided by the authors. The work is well organized and clearly written. The authors soundly investigate a worthwhile facet to this important disease and provide valuable new information on reptile immunogenetics. Results and conclusions are accurately presented and appropriately qualified where necessary. Prior points of review are adequately addressed in my opinion. My only minor critique is on lines 64-65 – causative associations between anthropogenic environmental change and immunosuppression have not been proven (and not by the cited references). They are associations at best (more often simply speculation) that fail to distinguish downstream effects of disease from predisposing conditions. Please rephrase this sentence.

Review form: Reviewer 2

Is the manuscript scientifically sound in its present form?

Yes

Are the interpretations and conclusions justified by the results?

Yes

Is the language acceptable?

Yes

Do you have any ethical concerns with this paper?

No

Have you any concerns about statistical analyses in this paper?

No

Recommendation?

Accept with minor revision (please list in comments)

Comments to the Author(s)

Please see attached pdf for comments (Appendix A).

Review form: Reviewer 3

Is the manuscript scientifically sound in its present form?

No

Are the interpretations and conclusions justified by the results?

No

Is the language acceptable?

Yes

Do you have any ethical concerns with this paper?

No

Have you any concerns about statistical analyses in this paper?

Yes

Recommendation?

Major revision is needed (please make suggestions in comments)

Comments to the Author(s)

General comments:

The MHC genetic results of the study are an important and welcome first step in understanding the immunogenetics behind FP in sea turtles. I disagree with the approach of classifying tumors based on visual appearance alone, and suggest that the categories of "active" and "regressing" FP be revised to only include turtles with recapture data showing tumors getting smaller and/or disappearing, and to exclude any turtles for which diagnosis was made on a single visual observation. Meaning the data will need to be re-analyzed for relative risk and any other analyses that include the two categories.

Alternatively, if you want to keep the categories as they are, you could change "regressing" to "smooth", and just characterize the tumors based on texture without making sweeping assumptions about disease progression based on appearance alone. With that major revision addressed, I think the paper will be greatly improved. More detailed comments are provided below.

Specific comments:

L26: change to "highlight" to agree with "associations" and "outcomes"

L49: suggest replace "with" with "containing"

L57: "chelonid alphaherpesvirus" should not be capitalized

L58: define *C. mydas* as "green sea turtles" at first mention

L59: FP has been described in all sea turtle species, not just *C. mydas* and *C. caretta*- please amend this sentence to reflect that, including pertinent references.

L60: delete "back"

L65-67: This summary overlooks a recent study on green turtle immunity and FP that should be included: <https://www.mdpi.com/2076-2615/11/3/861>

L77: change "green sea turtles" to "*C. mydas*" for consistency

L106-107: Need more details on venipuncture technique. How much blood was collected? Into what size tubes, and what kind of tubes? E.g., plain or anticoagulant, if anticoagulant then what kind? What length needles?

L107: Need more info on biopsy technique. Was aseptic technique used to prepare biopsies? If so, how was the biopsy site prepped? Any local anesthetic (e.g., lidocaine) used?

L111: suggest replace “diagnostic of” with “consistent with” ... a true diagnosis of FP requires confirmatory histopathology, preferably paired with a molecular diagnostic such as PCR. Visually observing tumors does not classify as “diagnosing”.

L111–112: I do not agree with this classification system, and I suggest that the tumor “active” and “regressing” categories be removed from the manuscript for all turtles for whom tumor size data upon repeat capture(s) did not support documentation of FP regression. In turn, I suggest removing all downstream analyses (e.g., relative risk) that were based on comparing active vs. regressed on visual inspection alone. Calling tumors “regressing” just because they have a smooth surface is a big stretch. FP tumors can be smooth, rugose, or some variation in between, and tumor texture does not always correlate with tumor progression. I have personally witnessed plenty of large and small tumors that had a smooth surface but were definitely not regressing- in fact, they were progressing in several cases. FP tumor characteristics including surface texture also vary by species, geographic region, and from which cell type(s) the tumor arises. Determining whether a tumor is in regression is a microscopic diagnosis, not a gross one, and it is not appropriate to make this assumption without proper evidence to support it. The reference cited here in fact seems to report the same error in some cases- diagnosing tumor regression based on surface contour alone- and does not actually provide scientific evidence that smooth FP tumors are in regression in every case. I recommend not perpetuating this unsupported distinction.

L113–115: What about if turtles were subsequently recaptured and tumors were smaller (but not absent)? Were those also categorized as “regressed”?

L119: I suggest clarifying here that the samples were “amplified using conventional polymerase chain reaction (cPCR)” ... (correct?)

Decision letter (RSOS-211190.R0)

Dear Dr Savage

The Editors assigned to your paper RSOS-211190 "Adaptive evolution of MHC class I immune genes and disease associations in coastal juvenile sea turtles" have now received comments from reviewers and would like you to revise the paper in accordance with the reviewer comments and any comments from the Editors. Please note this decision does not guarantee eventual acceptance.

All the reviewers are very positive about the paper and its findings. Some of the reviewers have raised a few relatively substantive points that will be important to carefully consider and respond to. We invite you to respond to the comments supplied below and revise your manuscript. Below the referees' and Editors' comments (where applicable) we provide additional requirements. Final acceptance of your manuscript is dependent on these requirements being met. We provide guidance below to help you prepare your revision.

Please submit your revised manuscript and required files (see below) no later than 21 days from today's (ie 17-Sep-2021) date. Note: the ScholarOne system will 'lock' if submission of the revision is attempted 21 or more days after the deadline. If you do not think you will be able to meet this deadline please contact the editorial office immediately.

on behalf of Professor Steve Brown (Associate Editor) and Steve Brown (Subject Editor)
openscience@royalsociety.org

Reviewer comments to Author:

Reviewer: 1

Comments to the Author(s)

Thank you for the opportunity to review "Adaptive evolution of MHC class I immune genes and disease associations in coastal juvenile sea turtles." In addition to my review of the current draft of the manuscript, I reviewed the comments of the two previous reviewers and the responses and edits provided by the authors. The work is well organized and clearly written. The authors soundly investigate a worthwhile facet to this important disease and provide valuable new information on reptile immunogenetics. Results and conclusions are accurately presented and appropriately qualified where necessary. Prior points of review are adequately addressed in my opinion. My only minor critique is on lines 64-65 – causative associations between anthropogenic environmental change and immunosuppression have not been proven (and not by the cited references). They are associations at best (more often simply speculation) that fail to distinguish downstream effects of disease from predisposing conditions. Please rephrase this sentence.

Reviewer: 2

Comments to the Author(s)

Please see attached pdf for comments.

Reviewer: 3

Comments to the Author(s)

General comments:

The MHC genetic results of the study are an important and welcome first step in understanding the immunogenetics behind FP in sea turtles. I disagree with the approach of classifying tumors based on visual appearance alone, and suggest that the categories of "active" and "regressing" FP

be revised to only include turtles with recapture data showing tumors getting smaller and/or disappearing, and to exclude any turtles for which diagnosis was made on a single visual observation. Meaning the data will need to be re-analyzed for relative risk and any other analyses that include the two categories.

Alternatively, if you want to keep the categories as they are, you could change "regressing" to "smooth", and just characterize the tumors based on texture without making sweeping assumptions about disease progression based on appearance alone. With that major revision addressed, I think the paper will be greatly improved. More detailed comments are provided below.

Specific comments:

L26: change to "highlight" to agree with "associations" and "outcomes"

L49: suggest replace "with" with "containing"

L57: "chelonid alphaherpesvirus" should not be capitalized

L58: define *C. mydas* as "green sea turtles" at first mention

L59: FP has been described in all sea turtle species, not just *C. mydas* and *C. caretta*- please amend this sentence to reflect that, including pertinent references.

L60: delete "back"

L65-67: This summary overlooks a recent study on green turtle immunity and FP that should be included: <https://www.mdpi.com/2076-2615/11/3/861>

L77: change "green sea turtles" to "*C. mydas*" for consistency

L106-107: Need more details on venipuncture technique. How much blood was collected? Into what size tubes, and what kind of tubes? E.g., plain or anticoagulant, if anticoagulant then what kind? What length needles?

L107: Need more info on biopsy technique. Was aseptic technique used to prepare biopsies? If so, how was the biopsy site prepped? Any local anesthetic (e.g., lidocaine) used?

L111: suggest replace "diagnostic of" with "consistent with"... a true diagnosis of FP requires confirmatory histopathology, preferably paired with a molecular diagnostic such as PCR. Visually observing tumors does not classify as "diagnosing".

L111-112: I do not agree with this classification system, and I suggest that the tumor "active" and "regressing" categories be removed from the manuscript for all turtles for whom tumor size data upon repeat capture(s) did not support documentation of FP regression. In turn, I suggest removing all downstream analyses (e.g., relative risk) that were based on comparing active vs. regressed on visual inspection alone. Calling tumors "regressing" just because they have a smooth surface is a big stretch. FP tumors can be smooth, rugose, or some variation in between, and tumor texture does not always correlate with tumor progression. I have personally witnessed plenty of large and small tumors that had a smooth surface but were definitely not regressing- in fact, they were progressing in several cases. FP tumor characteristics including surface texture also vary by species, geographic region, and from which cell type(s) the tumor arises. Determining whether a tumor is in regression is a microscopic diagnosis, not a gross one, and it is not appropriate to make this assumption without proper evidence to support it. The reference

cited here in fact seems to report the same error in some cases- diagnosing tumor regression based on surface contour alone- and does not actually provide scientific evidence that smooth FP tumors are in regression in every case. I recommend not perpetuating this unsupported distinction.

L113-115: What about if turtles were subsequently recaptured and tumors were smaller (but not absent)? Were those also categorized as "regressed"?

L119: I suggest clarifying here that the samples were "amplified using conventional polymerase chain reaction (cPCR)" ... (correct?)

===PREPARING YOUR MANUSCRIPT===

===PREPARING YOUR REVISION IN SCHOLARONE===

Author's Response to Decision Letter for (RSOS-211190.R0)

See Appendix B.

RSOS-211190.R1 (Revision)

Review form: Reviewer 2

Is the manuscript scientifically sound in its present form?

Yes

Are the interpretations and conclusions justified by the results?

No

Is the language acceptable?

Yes

Do you have any ethical concerns with this paper?

No

Have you any concerns about statistical analyses in this paper?

No

Recommendation?

Accept with minor revision (please list in comments)

Comments to the Author(s)

The authors have implemented all of the suggested changes (bar one, see below) to the manuscript, and expanded the requested sections. Therefore, if the remaining outstanding issue is satisfactorily addressed, I am happy to recommend the manuscript be accepted for publication in Royal Society Open Science.

Outstanding issues:

Remove “tumor regression” and “which may indicate tumor regression” from the abstract. The sentence should instead read “...two alleles significantly associated with smooth FP tumor texture.”. Reviewer 3 is correct, a smooth tumor cannot be assumed to be regressing or indicative of future regression. I, and other colleagues I have reached out to on this issue, have observed numerous smooth tumors aggressively growing. Smooth tumors should not be taken as a proxy for regression.

Throughout the revisions in the manuscript the authors continue to conflate tumor texture with regression status. Essentially stating ‘regressed/smooth tumor texture’ wherever they had previously said regressed. The authors need to remove all instances of regressed, regression etc., where not explicitly referring to recaptured turtles with confirmed regression, and refer only to texture instead. The authors should not continue to conflate smooth tumors with regression unless they can provide significant evidence that all smooth tumors regress. Reviewer 3 is correct that the cited paper does not provide adequate evidence that smooth tumor surface texture is indicative of regression. Again, observation by myself and colleagues at rehabilitation facilities confirm Reviewer 3’s assertion that smooth tumors are often actively growing, and that regression status cannot be ascertained by gross examination at a single time-point. Further evidence is required to determine whether any gross characterization can be significantly linked to current or future FP tumor regression. In the absence of such evidence, the authors should only refer to tumor texture through the manuscript, and refrain from implying that texture alone is a predictor of disease progression/regression status.

If the authors remove the unproven conflation between tumor texture and regression, stating instead only that which was empirically observable in their study cohort (texture), then I fully support the acceptance of this manuscript and believe it will make an important contribution to the field of sea turtle immunogenetics.

Review form: Reviewer 3

Is the manuscript scientifically sound in its present form?

No

Are the interpretations and conclusions justified by the results?

No

Is the language acceptable?

No

Do you have any ethical concerns with this paper?

No

Have you any concerns about statistical analyses in this paper?

Yes

Recommendation?

Major revision is needed (please make suggestions in comments)

Comments to the Author(s)

Methods & Results sections:

- Exactly how many turtles were recaptured with fewer/smaller/no tumors, indicating true FP regression? I cannot find this highly pertinent data in the manuscript, although the response letter from the authors states it to be N=7.

- Need to clarify whether "smooth" tumored turtles were lumped with those with true regression in the statistical analyses- if so, this is an artificial grouping based on weak/anecdotal data. I recommend that the few turtles with true regression be removed from this group and these statistics re-run to only include group with smooth tumors.

- Along those same lines, need to limit the speculative association between smooth tumors and FP regression to the introduction and discussion sections, and remove it from the methods and results sections completely. Instead, name the parameter "tumor texture" or "smooth" versus "verrucous", or something similar, and delete any mention of regression from these sections except for the small handful of cases in which true tumor regression was documented over multiple captures. This will be more accurate anyway once you separate out the turtles with smooth tumors and those with true regression.

Discussion section:

- In the Blackburn paper now cited reports that viral transcription was HIGHER in smooth tumors compared to verrucous ones, which suggests viral replication and tumor GROWTH, not regression which would be expected to be associated with little to no viral replication. Please rephrase/re-interpret.

Decision letter (RSOS-211190.R1)

Dear Dr Savage

The Editors assigned to your paper RSOS-211190.R1 "Adaptive evolution of MHC class I immune genes and disease associations in coastal juvenile sea turtles" have now received comments from reviewers and would like you to revise the paper in accordance with the reviewer comments and any comments from the Editors. Please note this decision does not guarantee eventual acceptance.

Both reviewers conclude that the manuscript is much improved, but require you to address further the issue of smooth tumors and regression. In particular, they both make various recommendations to further clarify and modify the text on the issue of smooth tumors as "regressed" and to address and remove references to the linked designation of "regression/texture". We invite you to respond to the comments supplied below and revise your manuscript. Below the referees' and Editors' comments (where applicable) we provide additional requirements. Final acceptance of your manuscript is dependent on these requirements being met. We provide guidance below to help you prepare your revision.

Please submit your revised manuscript and required files (see below) no later than 21 days from today's (ie 09-Nov-2021) date. Note: the ScholarOne system will 'lock' if submission of the revision is attempted 21 or more days after the deadline. If you do not think you will be able to meet this deadline please contact the editorial office immediately.

on behalf of Professor Steve Brown (Associate Editor) and Steve Brown (Subject Editor)
openscience@royalsociety.org

Reviewer comments to Author:

Reviewer: 3

Comments to the Author(s)

Methods & Results sections:

- Exactly how many turtles were recaptured with fewer/smaller/no tumors, indicating true FP regression? I cannot find this highly pertinent data in the manuscript, although the response letter from the authors states it to be N=7.

- Need to clarify whether "smooth" tumored turtles were lumped with those with true regression in the statistical analyses- if so, this is an artificial grouping based on weak/anecdotal data. I recommend that the few turtles with true regression be removed from this group and these statistics re-run to only include group with smooth tumors.

- Along those same lines, need to limit the speculative association between smooth tumors and FP regression to the introduction and discussion sections, and remove it from the methods and results sections completely. Instead, name the parameter "tumor texture" or "smooth" versus "verrucous", or something similar, and delete any mention of regression from these sections except for the small handful of cases in which true tumor regression was documented over multiple captures. This will be more accurate anyway once you separate out the turtles with smooth tumors and those with true regression.

Discussion section:

- In the Blackburn paper now cited reports that viral transcription was HIGHER in smooth tumors compared to verrucous ones, which suggests viral replication and tumor GROWTH, not regression which would be expected to be associated with little to no viral replication. Please rephrase/re-interpret.

Reviewer: 2

Comments to the Author(s)

The authors have implemented all of the suggested changes (bar one, see below) to the manuscript, and expanded the requested sections. Therefore, if the remaining outstanding issue is satisfactorily addressed, I am happy to recommend the manuscript be accepted for publication in Royal Society Open Science.

Outstanding issues:

Remove "tumor regression" and "which may indicate tumor regression" from the abstract. The sentence should instead read "...two alleles significantly associated with smooth FP tumor texture.". Reviewer 3 is correct, a smooth tumor cannot be assumed to be regressing or indicative of future regression. I, and other colleagues I have reached out to on this issue, have observed numerous smooth tumors aggressively growing. Smooth tumors should not be taken as a proxy for regression.

Throughout the revisions in the manuscript the authors continue to conflate tumor texture with regression status. Essentially stating 'regressed/smooth tumor texture' wherever they had previously said regressed. The authors need to remove all instances of regressed, regression etc., where not explicitly referring to recaptured turtles with confirmed regression, and refer only to texture instead. The authors should not continue to conflate smooth tumors with regression unless they can provide significant evidence that all smooth tumors regress. Reviewer 3 is correct that the cited paper does not provide adequate evidence that smooth tumor surface texture is indicative of regression. Again, observation by myself and colleagues at rehabilitation facilities confirm Reviewer 3's assertion that smooth tumors are often actively growing, and that regression status cannot be ascertained by gross examination at a single time-point. Further evidence is required to determine whether any gross characterization can be significantly linked

to current or future FP tumor regression. In the absence of such evidence, the authors should only refer to tumor texture through the manuscript, and refrain from implying that texture alone is a predictor of disease progression/regression status.

If the authors remove the unproven conflation between tumor texture and regression, stating instead only that which was empirically observable in their study cohort (texture), then I fully support the acceptance of this manuscript and believe it will make an important contribution to the field of sea turtle immunogenetics.

===PREPARING YOUR MANUSCRIPT===

If you have been asked to revise the written English in your submission as a condition of publication, you must do so, and you are expected to provide evidence that you have received language editing support. The journal would prefer that you use a professional language editing service and provide a certificate of editing, but a signed letter from a colleague who is a fluent speaker of English is acceptable. Note the journal has arranged a number of discounts for authors using professional language editing services (<https://royalsociety.org/journals/authors/benefits/language-editing/>).

===PREPARING YOUR REVISION IN SCHOLARONE===

Please ensure that you include a summary of your paper at Step 2 'Type, Title, & Abstract'. This should be no more than 100 words to explain to a non-scientific audience the key findings of your

research. This will be included in a weekly highlights email circulated by the Royal Society press office to national UK, international, and scientific news outlets to promote your work.

Author's Response to Decision Letter for (RSOS-211190.R1)

See Appendix C.

RSOS-211190.R2

Review form: Reviewer 2

Is the manuscript scientifically sound in its present form?

Yes

Are the interpretations and conclusions justified by the results?

Yes

Is the language acceptable?

Yes

Do you have any ethical concerns with this paper?

No

Have you any concerns about statistical analyses in this paper?

No

Recommendation?

Accept as is

Comments to the Author(s)

The authors have implemented all of the suggested changes to the manuscript (bar three minor oversights, see below), and have suitably adapted the discussion section to better reflect the uncertain relationship between FP tumor texture and disease outcome. Therefore, once the very minor corrections have been made, I am more than happy to recommend the manuscript be accepted for publication in Royal Society Open Science, as it provides important MHC class I diversity data for sea turtles.

Minor changes:

The word "regression/" should be removed from the following three places to reflect that the analysis now only compares tumor texture, and not the 7 confirmed cases of genuine tumor regression.

Line 230: "...n=99 for the tumor regression/texture model)."

Line 234: "For the final models predicting FP occurrence and tumor regression/texture,..."

Line 237: "For tumor regression/texture, the final model was built..."

Review form: Reviewer 3

Is the manuscript scientifically sound in its present form?

Yes

Are the interpretations and conclusions justified by the results?

Yes

Is the language acceptable?

Yes

Do you have any ethical concerns with this paper?

No

Have you any concerns about statistical analyses in this paper?

No

Recommendation?

Accept as is

Comments to the Author(s)

The authors have done a commendable job of addressing extensive reviewers' comments and the revision is now in good shape. I recommend that the manuscript is now acceptable for publication.

Decision letter (RSOS-211190.R2)

Dear Dr Savage

On behalf of the Editors, we are pleased to inform you that your Manuscript RSOS-211190.R2 "Adaptive evolution of MHC class I immune genes and disease associations in coastal juvenile sea turtles" has been accepted for publication in Royal Society Open Science subject to very minor revisions in accordance with the referees' reports - see comments of Reviewer 2 below. Please find the referees' comments along with any feedback from the Editors below my signature.

Please submit your revised manuscript and required files (see below) no later than 7 days from today's (ie 04-Jan-2022) date. Note: the ScholarOne system will 'lock' if submission of the revision is attempted 7 or more days after the deadline. If you do not think you will be able to meet this deadline please contact the editorial office immediately.

on behalf of Professor Steve Brown (Associate Editor) and Steve Brown (Subject Editor)
 openscience@royalsociety.org

Reviewer comments to Author:

Reviewer: 2

Comments to the Author(s)

The authors have implemented all of the suggested changes to the manuscript (bar three minor oversights, see below), and have suitably adapted the discussion section to better reflect the uncertain relationship between FP tumor texture and disease outcome. Therefore, once the very minor corrections have been made, I am more than happy to recommend the manuscript be accepted for publication in Royal Society Open Science, as it provides important MHC class I diversity data for sea turtles.

Minor changes:

The word "regression/" should be removed from the following three places to reflect that the analysis now only compares tumor texture, and not the 7 confirmed cases of genuine tumor regression.

Line 230: "...n=99 for the tumor regression/texture model)."

Line 234: "For the final models predicting FP occurrence and tumor regression/texture,..."

Line 237: "For tumor regression/texture, the final model was built..."

Reviewer: 3

Comments to the Author(s)

The authors have done a commendable job of addressing extensive reviewers' comments and the revision is now in good shape. I recommend that the manuscript is now acceptable for publication.

===PREPARING YOUR MANUSCRIPT===

one version should clearly identify all the changes that have been made (for instance, in coloured highlight, in bold text, or tracked changes);

Please ensure that you include an acknowledgements' section before your reference list/bibliography. This should acknowledge anyone who assisted with your work, but does not

qualify as an author per the guidelines at <https://royalsociety.org/journals/ethics-policies/openness/>.

===PREPARING YOUR REVISION IN SCHOLARONE===

- Ensure that your data access statement meets the requirements at <https://royalsociety.org/journals/authors/author-guidelines/#data>. You should ensure that you cite the dataset in your reference list. If you have deposited data etc in the Dryad repository, please only include the 'For publication' link at this stage. You should remove the 'For review' link.
- If you are requesting an article processing charge waiver, you must select the relevant waiver option (if requesting a discretionary waiver, the form should have been uploaded, see 'File upload' above).
- If you have uploaded any electronic supplementary (ESM) files, please ensure you follow the guidance at <https://royalsociety.org/journals/authors/author-guidelines/#supplementary-material> to include a suitable title and informative caption. An example of appropriate titling and captioning may be found at https://figshare.com/articles/Table_S2_from_Is_there_a_trade-off_between_peak_performance_and_performance_breadth_across_temperatures_for_aerobic_scope_in_teleost_fishes_/3843624.

Author's Response to Decision Letter for (RSOS-211190.R2)

See Appendix D.

Decision letter (RSOS-211190.R3)

Dear Dr Savage,

I am pleased to inform you that your manuscript entitled "Adaptive evolution of MHC class I immune genes and disease associations in coastal juvenile sea turtles" is now accepted for publication in Royal Society Open Science.

on behalf of Steve Brown (Subject Editor)
openscience@royalsociety.org

Appendix A

Reviewer's comments (RSOS-211190):

This is an important study of MHC class I diversity in sea turtles. The study provides important baseline data for green turtles, for which such diversity has not previously been assessed, and loggerheads at the Florida study sites, a novel geographic area for loggerhead MHC study. The authors reveal high rates of MHC diversity in both species, which unfortunately hampered their ability to identify any potentially strong associations between specific MHC alleles and FP tumor occurrence and regression.

The weakness of the manuscript was overinterpretation of the results, and statements not supported by current results. However, with the most recent round of reviews these issues have mostly been addressed.

The main issues remaining which still need to be addressed are i) deposition of the sequencing data in a public repository, ii) modifying the manuscript to more explicitly state the parameters of the study (i.e. not only reporting sample size and MHC fragment size in the methods) and iii) updating the manuscript to reflect recent advances identifying FP-associated gene expression changes in immune-related genes. Given the revealed MHC diversity and the lack of robust statistical association between MHC alleles and FP, the manuscript calls several times for future studies to investigate gene expression changes in both MHC and non-MHC immune genes. Yet, the manuscript fails to cite recent papers which have conducted such research, including a recent transcriptomics paper from the authors' own research group.

Given the novelty of the study in identifying MHC class I gene diversity in green turtles and the expansion of known loggerhead MHC alleles I would recommend the acceptance and publication of the manuscript in *Royal Society Open Science* upon completion of the revisions listed below.

Revisions:

Abstract, please change “Random forest modeling and risk ratio analysis of *C. mydas* alleles uncovered one allele consistently and significantly associated with FP tumor regression.”
to “Random forest modeling and risk ratio analysis of *C. mydas* alleles uncovered one allele significantly associated with FP tumor regression.”.

It is a stretch to claim this was a consistent association when the allele was only identified in four individuals. The veracity of the statistical significance is also uncertain given such a small sample size.

In the **Data Statement** the authors state that “The sequence data will be uploaded to GenBank upon acceptance.” This is not best practice. The data should be uploaded in advance of acceptance and the accession number explicitly provided in the manuscript. Additionally, advanced access to the deposited data should be provided to the Editor and Reviewers to confirm that the data has been correctly deposited.

GenBank offer the option to only make the data public after the associated paper has been released, but the data can and should be uploaded in advance of acceptance.

Discussion. In addition to assessing genetic diversity and immune gene expression changes, functional studies into the immune response in FP and non-FP afflicted turtles are important in understanding the ability of the immune system to suppress FP tumors. A sentence to this effect should be added to the Discussion section, and relevant literature cited, such as Sposato et al. 2021 ¹.

The manuscript calls several times for future studies to investigate gene expression changes in both MHC and non-MHC immune genes. Yet, the manuscript fails to cite recent papers which have conducted such research at each of these locations. The following papers which assessed FP gene expression changes, including immune gene focus, should be cited: Blackburn et al. 2021, Kane et al. 2021, Duffy et al. 2018, Yetsko et al. 2021 and Banerjee 2021 ²⁻⁶ (and any others relevant papers the authors identify). All of these papers identified differentially expressed immune-related genes in FP tumors, including ones linked to turtle outcome (Yetsko et al. 2021), CD3+ T lymphocyte infiltration in FP tumors (Yetsko et al. 2021) and as potential therapeutic avenues (Blackburn et al. 2021, immune checkpoint inhibitor associated differentially expressed genes).

Line 347 - 352: "...and/or immune gene expression patterns are important priorities..." and "While other studies have focused on the molecular evolution of ChHV5 [23] or its expression within tumors [80], evaluating the immunogenetic repertoire of sea turtles is an essential component of evaluating host-pathogen interactions underlying FP dynamics."

As above, the manuscript should not just refer to future studies. Published studies have already started this work and should be cited here. Please cite the following papers exploring immune-related gene expression and its correlation to tumor progression, outcome and ChHV5 in this sentence: Blackburn et al. 2021, Kane et al. 2021, Duffy et al. 2018, Yetsko et al. 2021 and Banerjee 2021, including the authors' own FP transcriptomics paper (Kane et al. 2021). Additionally, reference [80]⁴ profiled both host (including immune gene) and viral gene expression, and not just ChHV5 expression which is implied by the current format of this sentence.

Introduction, line 69 "Reptile immunogenetics is relatively understudied in comparison to other vertebrate taxa 70 [35]." I agree with the author's statement, but they should add a clarifying sentence that more recently immunogenetics-related research has been conducted in FP-afflicted green sea turtles, including citing Blackburn et al 2021 ² which had a focus on gene expression related to immune checkpoint inhibitors, Banerjee 2021 ⁶ which identified immune-response genes as being differentially expressed in the blood of sea turtles with and without FP, and Yetsko et al. 2021 ⁵ which reported changes in immune gene expression within FP tumors, including the expression of certain immune genes being correlated with the outcome of sea turtles with FP.

Line 365: "Gene expression studies could be especially illuminating in evaluating the role of MHC expression relative to tumor development and regression, where MHC expression may be a better predictor of FP status than presence or absence of specific alleles." Please cite existing FP gene expression studies in relation to this sentence, especially transcriptome-wide studies that investigated differentially expressed immune genes (i.e. Blackburn 2021; Kane 2021; Duffy 2018; Yetsko 2021 and Banerjee 2021).

Line 490: Please change “Reptilian immune systems are poorly studied relative to other vertebrate taxa [40], and the roles of the innate and acquired immune systems in fighting disease remain open questions in reptile biology [35].” to: “Reptilian immune systems are poorly studied relative to other vertebrate taxa [40], and although expression changes in immune-related genes have been consistently linked to FP tumor development and outcome (cite Blackburn 2021; Kane 2021; Duffy 2018; Yetsko 2021 and Banerjee 2021 here), the roles of the innate and acquired immune systems in fighting disease remain open questions in reptile biology [35].”

The authors should more prominently report the size of the gene fragment used for the analysis, not only reporting it in the Methods. Otherwise, readers may get a misleading impression of what proportion of the MHC was assessed.

The authors should clearly state in the abstract that the analysis was based on the sequencing of a 162bp fragment of MHC. It is important to explicitly state the fragment size as there may well be other alleles within non-sequenced regions of MHC genes which correlate to FP status.

Line 15: Please change “We sequenced the $\alpha 1$ peptide-binding region of MHC class I genes from juvenile green (*Chelonia mydas*) and loggerhead (*Caretta caretta*) sea turtles in Florida, USA.” to “We sequenced the $\alpha 1$ peptide-binding region of MHC class I genes (**162bp**) from **268** juvenile green (*Chelonia mydas*) and **88** loggerhead (*Caretta caretta*) sea turtles in Florida, USA.”

Change: “Figure 2: Bayesian phylogeny of 124 alleles of exon 2 of the I α major histocompatibility complex in...” to “Figure 2: Bayesian phylogeny of 124 alleles of exon 2 (**162bp**) of the I α major histocompatibility complex in...”

Change: “Figure 3: Haplotype network of 124 class I exon 2 α MHC alleles...” to “Figure 3: Haplotype network of 124 class I exon 2 α (**162bp**) MHC alleles...”

Recovering MHC alleles from 356 sea turtles (268 green and 88 loggerheads) is an impressive feat and key to defining the scope of the study. Therefore, in line 15 the authors should explicitly state the number of animals from which alleles were successfully sequenced. See line 15 changes above.

Line 307: Relative risk sub-section. While stated in the Methods, it should also be stated in this section of the Results that Chmy33 did not fall within the exclusion criteria of only retaining alleles that were found in ten or more individuals, but based on the author's perceived importance of this allele it was included even though it was only identified in 4 individuals. This is an important caveat of the analysis, which could be overlooked if only reported in the Methods and not the corresponding Results section.

Line 326: "The allele associated with FP occurrence, Chmy13..."

Earlier the authors state that Chmy13 is not significantly associated with FP occurrence after Bonferroni correction and this sentence should not claim an association. The sentence should therefore read "The allele **not significantly** associated with FP occurrence, Chmy13...",

Similarly for **line 327:** "In contrast, the alleles associated with FP regression, Chmy04 and Chmy33..." should read "In contrast, the alleles **weakly** associated with FP regression, Chmy04 and Chmy33...", as Chmy04 was not significantly associated after Bonferroni correction, and while Chmy33 had significance, it was only included after undergoing special exclusion from the originally defined inclusion criteria and was only present in 4 green turtles out of 268 green turtles in the study, raising the possibility that it is a statistical artifact.

Line 866: Please change: "**Figure 4:** Mean decrease in accuracy and Gini impurity..." to "**Figure 4:** Mean decrease in accuracy (left) and Gini impurity (right)..."

Line 869: Please change "**Figure 5:** Mean decrease in accuracy and Gini impurity..." to "**Figure 5:** Mean decrease in accuracy (left) and Gini impurity (right)..."

References:

1. Sposato, P., Keating, P., Lutz, P.L. & Milton, S.L. Evaluation of Immune Function In Two Populations of Green Sea Turtles (*Chelonia mydas*) In A Degraded Versus A Nondegraded Habitat. *Journal of Wildlife Diseases* (2021).
2. Blackburn, N.B. *et al.* Transcriptomic Profiling of Fibropapillomatosis in Green Sea Turtles (*Chelonia mydas*) From South Texas. *Frontiers in Immunology* **12**(2021).
3. Kane, R.A. *et al.* Gene expression changes with tumor disease and leech parasitism in the juvenile green sea turtle skin transcriptome. *Gene*, 145800 (2021).

4. Duffy, D.J. *et al.* Sea turtle fibropapilloma tumors share genomic drivers and therapeutic vulnerabilities with human cancers. *Communications Biology* **1**, 63 (2018).
5. Yetsko, K. *et al.* Molecular characterization of a marine turtle tumor epizootic, profiling external, internal and postsurgical regrowth tumors. *Communications Biology* **4**, 152 (2021).
6. Banerjee, S.M. *et al.* Species and population specific gene expression in blood transcriptomes of marine turtles. *BMC Genomics* **22**, 346 (2021).

Appendix B

Response to reviews for RSOS-211190, titled Adaptive evolution of MHC class I immune genes and disease associations in coastal juvenile sea turtles.

Please find our responses to all reviewer comments below. We have addressed all suggested modifications and our responses are in bold italicized text and line numbers refer to those in the track-changed document.

Note to editor and reviewers:

*In exploring the suggestions of reviewer #2, we found an error where MHC supertypes were not accurately tallied per turtle in our dataframe. The supertype error affected only the random forest analyses (the classification of supertypes via *dapc* was not affected, just the tally of supertypes present in each individual). Thus, we corrected the data and re-ran the analyses, and the interpretations of the results remain the same: our random forest models have moderate accuracy and out of box error, and the same alleles are weakly associated with an increased risk of FP or FP texture (e.g., *Chmy33*).*

We also ran risk ratio analyses for FP status and tumor texture as it relates to MHC supertypes, and have updated the methods and results. We had not previously report these analyses because the results were not remotely significant. In running the corrected dataframe, our chief finding from this new analysis is that MHC supertype B is associated with a slightly decreased risk of FP but after Bonferroni correction this association was not significant (results section, lines 349-353; supplemental figures S7 and S8). Our interpretation and discussion of MHC's role in FP dynamics remain largely unchanged, but the slight changes based on corrected results are now presented and discussed.

*Additionally, we found one individual in our dataset that was incorrectly coded as being FP-free when in fact it had FP tumors. Additionally, this individual was recaptured later without any visible external tumors and so was considered a truly regressed individual. We corrected this prior to re-running analyses, and it resulted in only slight changes to the random forest models and risk ratio analyses. Namely, allele *Chmy22*, which this individual had, also increased in importance in the tumor texture random forest analysis which prompted us to include *Chmy22* in the FP texture risk ratio analysis. After Bonferroni correction, it still was significantly associated with increased risk of smooth texture tumors. Even with this updated dataframe and results, there were no changes to the qualitative interpretation of either analysis or our discussion of the results.*

Reviewer 1

Thank you for the opportunity to review “Adaptive evolution of MHC class I immune genes and disease associations in coastal juvenile sea turtles.” In addition to my review of the current draft of the manuscript, I reviewed the comments of the two previous reviewers and the responses and edits provided by the authors. The work is well organized and clearly written. The authors soundly investigate a worthwhile facet to this important disease and provide valuable new information on reptile immunogenetics. Results and conclusions are accurately presented and appropriately qualified where necessary. Prior points of review are adequately addressed in my opinion. My only minor critique is on lines 64-65 – causative associations between anthropogenic environmental change and immunosuppression have not been proven (and not by the cited references). They are associations at best (more often simply speculation) that fail to

distinguish downstream effects of disease from predisposing conditions. Please rephrase this sentence.

Response: We thank the reviewer for their feedback, and we have re-worded this sentence to dissociate FP-related immunosuppression from environmental change (lines 66-69).

Reviewer 2

Reviewer's comments (RSOS-211190):

This is an important study of MHC class I diversity in sea turtles. The study provides important baseline data for green turtles, for which such diversity has not previously been assessed, and loggerheads at the Florida study sites, a novel geographic area for loggerhead MHC study. The authors reveal high rates of MHC diversity in both species, which unfortunately hampered their ability to identify any potentially strong associations between specific MHC alleles and FP tumor occurrence and regression.

The weakness of the manuscript was overinterpretation of the results, and statements not supported by current results. However, with the most recent round of reviews these issues have mostly been addressed. The main issues remaining which still need to be addressed are i) deposition of the sequencing data in a public repository, ii) modifying the manuscript to more explicitly state the parameters of the study (i.e. not only reporting sample size and MHC fragment size in the methods) and iii) updating the manuscript to reflect recent advances identifying FP-associated gene expression changes in immune-related genes. Given the revealed MHC diversity and the lack of robust statistical association between MHC alleles and FP, the manuscript calls several times for future studies to investigate gene expression changes in both MHC and non-MHC immune genes. Yet, the manuscript fails to cite recent papers which have conducted such research, including a recent transcriptomics paper from the authors' own research group.

Given the novelty of the study in identifying MHC class I gene diversity in green turtles and the expansion of known loggerhead MHC alleles I would recommend the acceptance and publication of the manuscript in *Royal Society Open Science* upon completion of the revisions listed below.

Response: We thank the reviewer for these helpful comments. Adding recent studies on gene expression better contextualizes our discussion and we appreciate the reviewer compiling them for us in their comments. We have updated the text to include these citations, as well as to add details on the sample size and sequencing parameters in the abstract. As of resubmission, we have also initiated the public release of our sequences on GenBank (final release by NCBI staff is anticipated shortly). More details below.

Revisions:

Abstract, please change “Random forest modeling and risk ratio analysis of *C. mydas* alleles uncovered one allele consistently and significantly associated with FP tumor regression.”

to “Random forest modeling and risk ratio analysis of *C. mydas* alleles uncovered one allele significantly associated with FP tumor regression.”.

It is a stretch to claim this was a consistent association when the allele was only identified in four individuals. The veracity of the statistical significance is also uncertain given such a small sample size.

Response: *We have updated the text accordingly (line 21).*

In the **Data Statement** the authors state that “The sequence data will be uploaded to GenBank upon acceptance.” This is not best practice. The data should be uploaded in advance of acceptance and the accession number explicitly provided in the manuscript. Additionally, advanced access to the deposited data should be provided to the Editor and Reviewers to confirm that the data has been correctly deposited.

GenBank offer the option to only make the data public after the associated paper has been released, but the data can and should be uploaded in advance of acceptance.

Response: *We have uploaded the data to GenBank and have initiated its public release as of resubmission of this manuscript. The GenBank numbers have also been added in the data statement.*

Discussion. In addition to assessing genetic diversity and immune gene expression changes, functional studies into the immune response in FP and non-FP afflicted turtles are important in understanding the ability of the immune system to suppress FP tumors. A sentence to this effect should be added to the Discussion section, and relevant literature cited, such as Sposato et al. 2021¹.

The manuscript calls several times for future studies to investigate gene expression changes in both MHC and non-MHC immune genes. Yet, the manuscript fails to cite recent papers which have conducted such research at each of these locations. The following papers which assessed FP gene expression changes, including immune gene focus, should be cited: Blackburn et al. 2021, Kane et al. 2021, Duffy et al. 2018, Yetsko et al. 2021 and Banerjee 2021²⁻⁶ (and any others relevant papers the authors identify). All of these papers identified differentially expressed immune-related genes in FP tumors, including ones linked to turtle outcome (Yetsko et al. 2021), CD3+ T lymphocyte infiltration in FP tumors (Yetsko et al. 2021) and as potential therapeutic avenues (Blackburn et al. 2021, immune checkpoint inhibitor associated differentially expressed genes).

Response: *We appreciate these specific suggestions on how to incorporate gene expression studies into our discussion and we now cite them throughout.*

Line 347 - 352: “...and/or immune gene expression patterns are important priorities...” and “While other studies have focused on the molecular evolution of ChHV5 [23] or its expression within tumors [80], evaluating the immunogenetic repertoire of sea turtles is an essential component of evaluating host-pathogen interactions underlying FP dynamics.”

As above, the manuscript should not just refer to future studies. Published studies have already started this work and should be cited here. Please cite the following papers exploring immune-

related gene expression and its correlation to tumor progression, outcome and ChHV5 in this sentence: Blackburn et al. 2021, Kane et al. 2021, Duffy et al. 2018, Yetsko et al. 2021 and Banerjee 2021, including the authors' own FP transcriptomics paper (Kane et al. 2021).

Additionally, reference [80]⁴ profiled both host (including immune gene) and viral gene expression, and not just ChHV5 expression which is implied by the current format of this sentence.

Response: We have updated the first paragraph of the discussion to thoroughly reference these studies and their utility in understanding sea turtle immune systems and FP response (lines 393-403).

Introduction, line 69 “Reptile immunogenetics is relatively understudied in comparison to other vertebrate taxa 70 [35].” I agree with the author’s statement, but they should add a clarifying sentence that more recently immunogenetics-related research has been conducted in FP-afflicted green sea turtles, including citing Blackburn et al 2021² which had a focus on gene expression related to immune checkpoint inhibitors, Banerjee 2021⁶ which identified immune-response genes as being differentially expressed in the blood of sea turtles with and without FP, and Yetsko et al. 2021⁵ which reported changes in immune gene expression within FP tumors, including the expression of certain immune genes being correlated with the outcome of sea turtles with FP.

Response: We appreciate this suggestion and have added these citations to an earlier sentence in the preceding paragraph summarizing current immune studies in sea turtles, where we also cite two recent studies (Perrault et al. 2021 and Sposato et al. 2021) on immune function in FP-afflicted C. mydas (lines 71-75). We have removed the sentence “Reptile immunogenetics is relatively understudied in comparison to other vertebrate taxa” from the start of fourth introduction paragraph.

While Banerjee et al. 2021 does not explicitly analyze FP-positive turtles, we have referenced this study in our discussion about the broad importance of transcriptome studies in understanding sea turtle physiology and immune response (line 427-431)

Line 365: “Gene expression studies could be especially illuminating in evaluating the role of MHC expression relative to tumor development and regression, where MHC expression may be a better predictor of FP status than presence or absence of specific alleles.” Please cite existing FP gene expression studies in relation to this sentence, especially transcriptome-wide studies that investigated differentially expressed immune genes (i.e. Blackburn 2021; Kane 2021; Duffy 2018; Yetsko 2021 and Banerjee 2021).

Response: We have updated this sentence to include these citations (lines 427-431, “Based on previous transcriptome-wide studies in sea turtles...”)

Line 490: Please change “Reptilian immune systems are poorly studied relative to other vertebrate taxa [40], and the roles of the innate and acquired immune systems in fighting disease remain open questions in reptile biology [35].” to: “Reptilian immune systems are poorly studied

relative to other vertebrate taxa [40], and although expression changes in immune-related genes have been consistently linked to FP tumor development and outcome (cite Blackburn 2021; Kane 2021; Duffy 2018; Yetsko 2021 and Banerjee 2021 here), the roles of the innate and acquired immune systems in fighting disease remain open questions in reptile biology [35].”

Response: *We have updated this sentence as suggested by the reviewer (lines 555-556) but have not included the Banerjee reference in this sentence, as it does not explicitly analyze FP vs. non-FP turtles (but the paper is cited elsewhere in discussion, line 427-431)*

The authors should more prominently report the size of the gene fragment used for the analysis, not only reporting it in the Methods. Otherwise, readers may get a misleading impression of what proportion of the MHC was assessed. The authors should clearly state in the abstract that the analysis was based on the sequencing of a 162bp fragment of MHC. It is important to explicitly state the fragment size as there may well be other alleles within non-sequenced regions of MHC genes which correlate to FP status.

Line 15: Please change “We sequenced the $\alpha 1$ peptide-binding region of MHC class I genes from juvenile green (*Chelonia mydas*) and loggerhead (*Caretta caretta*) sea turtles in Florida, USA.” to “We sequenced the $\alpha 1$ peptide-binding region of MHC class I genes (**162bp**) from **268** juvenile green (*Chelonia mydas*) and **88** loggerhead (*Caretta caretta*) sea turtles in Florida, USA.”

Response: *We have changed the text as suggested (lines 15-16)*

Change: “Figure 2: Bayesian phylogeny of 124 alleles of exon 2 of the I α major histocompatibility complex in...” to “Figure 2: Bayesian phylogeny of 124 alleles of exon 2 (**162bp**) of the I α major histocompatibility complex in...”

Response: *We have changed the caption as suggested.*

Change: “Figure 3: Haplotype network of 124 class I exon 2 α MHC alleles...” to “Figure 3: Haplotype network of 124 class I exon 2 α (**162bp**) MHC alleles...”

Response: *We have changed the caption as suggested.*

Recovering MHC alleles from 356 sea turtles (268 green and 88 loggerheads) is an impressive feat and key to defining the scope of the study. Therefore, in line 15 the authors should explicitly state the number of animals from which alleles were successfully sequenced. See line 15 changes above.

Response: *We appreciate the reviewer pointing this out and have changed the text accordingly.*

Line 307: Relative risk sub-section. While stated in the Methods, it should also be stated in this section of the Results that Chmy33 did not fall within the exclusion criteria of only retaining alleles that were found in ten or more individuals, but based on the author’s perceived importance of this allele it was included even though it was only identified in 4 individuals. This is an important caveat of the analysis, which could be overlooked if only reported in the Methods and not the corresponding Results section.

Response: We now include this caveat explicitly in the results section (lines 343-346)

Line 326: “The allele associated with FP occurrence, Chmy13...” Earlier the authors state that Chmy13 is not significantly associated with FP occurrence after Bonferroni correction and this sentence should not claim an association. The sentence should therefore read “The allele **not significantly** associated with FP occurrence, Chmy13...”

Response: We have changed the text as suggested (lines 365-365)

Similarly for **line 327:** “In contrast, the alleles associated with FP regression, Chmy04 and Chmy33...” should read “In contrast, the alleles **weakly** associated with FP regression, Chmy04 and Chmy33...”, as Chmy04 was not significantly associated after Bonferroni correction, and while Chmy33 had significance, it was only included after undergoing special exclusion from the originally defined inclusion criteria and was only present in 4 green turtles out of 268 green turtles in the study, raising the possibility that it is a statistical artifact.

Response: We have changed the text as suggested (line 366).

Line 866: Please change: “**Figure 4:** Mean decrease in accuracy and Gini impurity...” to “**Figure 4:** Mean decrease in accuracy (left) and Gini impurity (right)...”

Response: We have changed the caption as suggested.

Line 869: Please change “**Figure 5:** Mean decrease in accuracy and Gini impurity...” to “**Figure 5:** Mean decrease in accuracy (left) and Gini impurity (right)...”

Response: We have changed the caption as suggested.

Reviewer 3

General comments:

The MHC genetic results of the study are an important and welcome first step in understanding the immunogenetics behind FP in sea turtles. **I disagree with the approach of classifying tumors based on visual appearance alone, and suggest that the categories of "active" and "regressing" FP be revised to only include turtles with recapture data showing tumors getting smaller and/or disappearing, and to exclude any turtles for which diagnosis was made on a single visual observation. Meaning the data will need to be re-analyzed for relative risk and any other analyses that include the two categories.**

Alternatively, if you want to keep the categories as they are, you could change "regressing" to "smooth", **and just characterize the tumors based on texture without making sweeping assumptions about disease progression based on appearance alone. With that major revision addressed, I think the paper will be greatly improved.** More detailed comments are provided below.

Response: We appreciate the reviewer’s constructive comments and suggestions concerning tumor texture and analysis. Because too few of our turtles (n = 7) were categorized as regressed based on recapture data to run statistical analyses, we thus needed to retain the

turtles characterized based on tumor texture. We have re-analyzed the tumors according to the categories of “rough” and “smooth” as suggested rather than inferring regression based on tumor texture (specific line numbers below). We acknowledge in both the methods (lines 128-130) and in the discussion (lines 379-383, that while tumor texture and appearance is not definitively linked to disease outcome, there is evidence from a 2007 study done in two of the field sites sampled in our study that smooth tumors are likely regressed (Hirama and Ehrhart 2007, reference 34 in our manuscript), and our preliminary investigation illustrates a weak association between two MHC alleles (Chmy33 and Chmy22) and smooth tumor texture/regressed recaptured turtles. Coupled with a recent study (Blackburn et al. 2021) that reports smooth-textured tumors had higher levels of viral transcription, we suggest in our discussion that tumor appearance/texture and its relation to disease severity is a potential area for future studies (lines 382-383).

Specific comments:

L26: change to “highlight” to agree with “associations” and “outcomes”

Response: We have changed this wording (line 27)

L49: suggest replace “with” with “containing”

Response: We have changed this wording (line 47)

L57: “chelonid alphaherpesvirus” should not be capitalized

Response: We have changed this to sentence case (line 58)

L58: define *C. mydas* as “green sea turtles” at first mention

Response: Changed as suggested (lines 60)

L59: FP has been described in all sea turtle species, not just *C. mydas* and *C. caretta*- please amend this sentence to reflect that, including pertinent references.

Response: We have clarified that FP is documented in all species of sea turtles and added references for each (line 60)

L60: delete “back”

Response: Changed as suggested (line 62)

L65–67: This summary overlooks a recent study on green turtle immunity and FP that should be included: <https://www.mdpi.com/2076-2615/11/3/861>

Response: We appreciate the reviewer pointing out this recent study and have included Perrault et al. 2021 as well as Sposato et al. 2021 to demonstrate work being done on immune function in turtles with and without FP (lines 72-75).

L77: change “green sea turtles” to “*C. mydas*” for consistency

Response: *Changed as suggested.*

L106–107: Need more details on venipuncture technique. How much blood was collected? Into what size tubes, and what kind of tubes? E.g., plain or anticoagulant, if anticoagulant then what kind? What length needles?

Response: *We have updated the text to include details on venipuncture and blood collection (lines 117-120).*

L107: Need more info on biopsy technique. Was aseptic technique used to prepare biopsies? If so, how was the biopsy site prepped? Any local anesthetic (e.g., lidocaine) used?

Response: *We have updated this section to include information on biopsy technique (lines 120-123)*

L111: suggest replace “diagnostic of” with “consistent with”... a true diagnosis of FP requires confirmatory histopathology, preferably paired with a molecular diagnostic such as PCR. Visually observing tumors does not classify as “diagnosing”.

Response: *Changed as suggested (line 126)*

L111–112: I do not agree with this classification system, and I suggest that the tumor “active” and “regressing” categories be removed from the manuscript **for all turtles for whom tumor size data upon repeat capture(s) did not support documentation of FP regression**. In turn, I suggest removing all downstream analyses (e.g., relative risk) that were based on comparing active vs. regressed on visual inspection alone. Calling tumors “regressing” just because they have a smooth surface is a big stretch. FP tumors can be smooth, rugose, or some variation in between, and tumor texture does not always correlate with tumor progression. I have personally witnessed plenty of large and small tumors that had a smooth surface but were definitely not regressing- in fact, they were progressing in several cases. FP tumor characteristics including surface texture also vary by species, geographic region, and from which cell type(s) the tumor arises. Determining whether a tumor is in regression is a microscopic diagnosis, not a gross one, and it is not appropriate to make this assumption without proper evidence to support it. The reference cited here in fact seems to report the same error in some cases- diagnosing tumor regression based on surface contour alone- and does not actually provide scientific evidence that smooth FP tumors are in regression in every case. **I recommend not perpetuating this unsupported distinction.**

Response: *We appreciate the discussion on this important matter. As only seven of our individuals were categorized as regressed based on the disappearance of external tumors from first capture to subsequent recapture, we lack the recapture data to run statistical analyses and we thus needed to retain the turtles characterized based on tumor texture. We have re-analyzed the tumors according to the categories of “rough” and “smooth” as suggested rather*

than inferring regression based on tumor texture (lines 126-132 in methods) and have acknowledged the caveats preventing us from definitively inferring disease outcome from tumor appearance and texture. We are unaware of any published studies that find evidence that smooth tumors are not indicative of regression, and we would appreciate any pertinent citations from this reviewer to include in our discussion section if they are aware of any.

L113–115: What about if turtles were subsequently recaptured and tumors were smaller (but not absent)? Were those also categorized as "regressed"?

Response: Data pertaining to tumor size was not readily available, and so change in tumor size was not used to infer regression.

L119: I suggest clarifying here that the samples were "amplified using conventional polymerase chain reaction (cPCR)" ... (correct?)

Response: Correct, and we have revised the text accordingly (line 138).

Appendix C

Response to reviews for RSOS-211190, titled Adaptive evolution of MHC class I immune genes and disease associations in coastal juvenile sea turtles.

We would like to thank the editors and reviewers for their insight and advice. Please find our responses to all reviewer comments below. We have addressed all suggested modifications and our responses are in bold text and line numbers refer to those in the track-changed document.

Reviewer: 3

Comments to the Author(s)

Methods & Results sections:

- Exactly how many turtles were recaptured with fewer/smaller/no tumors, indicating true FP regression? I cannot find this highly pertinent data in the manuscript, although the response letter from the authors states it to be $N=7$.

Response: We have added this information to the methods section (lines 126-128)

- Need to clarify whether "smooth" tumored turtles were lumped with those with true regression in the statistical analyses- if so, this is an artificial grouping based on weak/anecdotal data. I recommend that the few turtles with true regression be removed from this group and these statistics re-run to only include group with smooth tumors.

Response: As suggested, we have removed from the random forest and risk ratio analyses the 7 truly regressed individuals and have re-run only using the group with smooth tumors ($n = 99$ *C. mydas*). The random forest and relative risk methods sections now explicitly state that they were only run on the 99 smooth-textured individuals (lines 224-225; 252-253).

- Along those same lines, need to limit the speculative association between smooth tumors and FP regression to the introduction and discussion sections, and remove it from the methods and results sections completely. Instead, name the parameter "tumor texture" or "smooth" versus "verrucous", or something similar, and delete any mention of regression from these sections except for the small handful of cases in which true tumor regression was documented over multiple captures. This will be more accurate anyway once you separate out the turtles with smooth tumors and those with true regression.

Response: We have removed mention of FP regression and smooth tumors from the methods and results (see line numbers below) to limit our speculation.

We would like to note that the results of the random forest model for tumor texture changed slightly; the out of bag error and model accuracy remained similar, but alleles Chmy33 and Chmy22 no longer had high model importance. This may be in part because, of the 7 truly regressed individuals removed from analyses, one had Chmy33 and one had Chmy22. As such, Chmy33 and Chmy22 were removed from subsequent relative risk analyses (their previous inclusion was predicated on their importance in the random forest model).

Methods:

- **Lines 120-126**
- **Lines 215-216**
- **Line 222**
- **Lines 224-227**
- **Lines 250-253**

Results:

- **Figure 1 and caption/phylogeny corrected**
- **Figure 5 and caption/random forest of texture**
- **Random forest and relative risk section wording and results updated to reflect new analyses with smooth-textured tumor individuals only (lines 317-348)**

Discussion section:

- In the Blackburn paper now cited reports that viral transcription was HIGHER in smooth tumors compared to verrucous ones, which suggests viral replication and tumor GROWTH, not regression which would be expected to be associated with little to no viral replication. Please rephrase/re-interpret.

Response: We have now explicitly stated that higher levels of viral transcripts suggests tumor growth (line 418-420)

Reviewer: 2

Comments to the Author(s)

The authors have implemented all of the suggested changes (bar one, see below) to the manuscript, and expanded the requested sections. Therefore, if the remaining outstanding issue is satisfactorily addressed, I am happy to recommend the manuscript be accepted for publication in Royal Society Open Science.

Outstanding issues:

Remove “tumor regression” and “which may indicate tumor regression” from the abstract. The sentence should instead read “...two alleles significantly associated with smooth FP tumor texture.”. Reviewer 3 is correct, a smooth tumor cannot be assumed to be regressing or indicative of future regression. I, and other colleagues I have reached out to on this issue, have observed numerous smooth tumors aggressively growing. Smooth tumors should not be taken as a proxy for regression.

Throughout the revisions in the manuscript the authors continue to conflate tumor texture with regression status. Essentially stating ‘regressed/smooth tumor texture’ wherever they had previously said regressed. The authors need to remove all instances of regressed, regression etc., where not explicitly referring to recaptured turtles with confirmed regression, and refer only to texture instead. The authors should not continue to conflate smooth tumors with regression unless they can provide significant evidence that all

smooth tumors regress. Reviewer 3 is correct that the cited paper does not provide adequate evidence that smooth tumor surface texture is indicative of regression. Again, observation by myself and colleagues at rehabilitation facilities confirm Reviewer 3's assertion that smooth tumors are often actively growing, and that regression status cannot be ascertained by gross examination at a single time-point. Further evidence is required to determine whether any gross characterization can be significantly linked to current or future FP tumor regression. In the absence of such evidence, the authors should only refer to tumor texture through the manuscript, and refrain from implying that texture alone is a predictor of disease progression/regression status.

If the authors remove the unproven conflation between tumor texture and regression, stating instead only that which was empirically observable in their study cohort (texture), then I fully support the acceptance of this manuscript and believe it will make an important contribution to the field of sea turtle immunogenetics.

Response: We thank Reviewer #2 for their advice and for their support of our manuscript.

Per the suggestion of reviewer #3, we have edited the methods and results to remove the conflation between regression and texture (line numbers listed above) but have retained some interpretation in the discussion section to provide context for why tumor texture was a focus for our study and why it may be an area for future research into FP disease progression (lines 420-427). In the abstract, we have changed the wording to not directly conflate smooth tumor texture with regression and now state that texture may be associated with disease outcome (line 22) In the discussion, we explicitly mention that the Hirama & Ehrhart 2007 paper is only an anecdotal link between tumor texture and progression/regression (lines 420-422) and we also cite Blackburn et al. 2021 to highlight a possible alternative relationship-- that smooth tumor texture may be associated with tumor growth (lines 418-420; 422-427), We believe that the tumor texture observations that reviewers 2 and 3 and their colleagues have made in a rehabilitation setting are extremely important to the FP field, and we look forward to their publication so that they can be properly cited.

Appendix D

Response to reviews for RSOS-211190, titled Adaptive evolution of MHC class I immune genes and disease associations in coastal juvenile sea turtles.

We would like to thank the editors and the reviewers for all of their feedback and insight throughout this review process, as we believe it has greatly strengthened our manuscript. We have addressed all suggested modifications and our responses are in bold text and line numbers refer to those in the track-changed document.

Reviewer: 2

Comments to the Author(s)

The authors have implemented all of the suggested changes to the manuscript (bar three minor oversights, see below), and have suitably adapted the discussion section to better reflect the uncertain relationship between FP tumor texture and disease outcome. Therefore, once the very minor corrections have been made, I am more than happy to recommend the manuscript be accepted for publication in Royal Society Open Science, as it provides important MHC class I diversity data for sea turtles.

We appreciate this reviewer's time and effort throughout multiple rounds of revision to make our manuscript better.

Minor changes:

The word "regression/" should be removed from the following three places to reflect that the analysis now only compares tumor texture, and not the 7 confirmed cases of genuine tumor regression.

Line 230: "...n=99 for the tumor regression/texture model)."

Line 234: "For the final models predicting FP occurrence and tumor regression/texture,..."

Line 237: "For tumor regression/texture, the final model was built..."

We have removed "regression/" from all of these instances (lines 223, 227, and 230 in track changed document).

Reviewer: 3

Comments to the Author(s)

The authors have done a commendable job of addressing extensive reviewers' comments and the revision is now in good shape. I recommend that the manuscript is now acceptable for publication.

We thank this reviewer for their thorough feedback throughout multiple revisions to improve this manuscript.